# Interactive analysis of single-cell epigenomic landscapes with ChromSCape

Pacôme Prompsy [1,2 ✉], Pia Kirchmeier[1,2], Justine Marsolier [1,2], Marc Deloger [3], Nicolas Servant [3] &
Céline Vallot [1,2 ✉]

Chromatin modifications orchestrate the dynamic regulation of gene expression during development and in disease. Bulk approaches have characterized the wide repertoire of histone modifications across cell types, detailing their role in shaping cell identity. However, these population-based methods do not capture cell-to-cell heterogeneity of chromatin landscapes, limiting our appreciation of the role of chromatin in dynamic biological processes. Recent technological developments enable the mapping of histone marks at single-cell resolution, opening up perspectives to characterize the heterogeneity of chromatin marks in complex biological systems over time. Yet, existing tools used to analyze bulk histone modifications profiles are not fit for the low coverage and sparsity of single-cell epigenomic datasets. Here, we present ChromSCape, a user-friendly interactive Shiny/R application distributed as a Bioconductor package, that processes single-cell epigenomic data to assist the biological interpretation of chromatin landscapes within cell populations. ChromSCape analyses the distribution of repressive and active histone modifications as well as chromatin accessibility landscapes from single-cell datasets. Using ChromSCape, we deconvolve chromatin landscapes within the tumor micro-environment, identifying distinct H3K27me3 landscapes associated with cell identity and breast tumor subtype.

[1] CNRS UMR3244, Institut Curie, PSL Research University, 26 rue d'Ulm, 75005 Paris, France. [2] Translational Research Department, Institut Curie, PSL Research University, 26 rue d'Ulm, 75005 Paris, France. [3] INSERM U900, Institut Curie, PSL Research University, Mines ParisTech, 26 rue d'Ulm, 75005 Paris, France. ✉email: pacome.prompsy@curie.fr; celine.vallot@curie.fr

Histone modifications are key regulators of gene expression, driving chromatin folding, and gene accessibility to transcription machineries. The recent development of single-cell methods to study epigenomes now enables the appreciation of the heterogeneity of chromatin modifications within a population. These experimental methods assess the distribution of histone marks at single-cell resolution by coupling next-generation sequencing to high-throughput microfluidics DNA barcoding (scChIP-seq)[1,2] or in situ reactions (scChIL-seq[3], scChIC-seq[4], scCUT&Tag[5]). In contrast to scATAC-seq approaches that identify open regions of the chromatin[6–8], these methods can capture various chromatin states, enriched in repressive or active histone marks (H3K27me3 or H3K4me3 for example). Using these approaches, we can study the heterogeneity of epigenomes within complex biological samples, such as tumors[1], and start appreciating the role of epigenomic diversity and the dynamics of chromatin in disease and development.

Existing tools used to analyze bulk ChIP-seq experiments are not fit for the low coverage and sparsity of these single-cell histone modifications datasets, which is due to the inherent low number of copy of DNA molecules per cell—maximum two for a diploid genome. Several computational methods for the analysis of scATAC-seq have been developed to deal with the specificities of single-cell DNA-based datasets. They were recently benchmarked[9], with SnapATAC[10], CisTopic[11], and Cusanovich2018[12] being the top-three performing methods. These tools, initially dedicated to scATAC-seq and without graphic interface, require some scripting skills. Biologists with limited computational training can manipulate and analyze scRNA-seq and scATAC-seq datasets using applications such as "scOrange"[13] and "'SCRAT"[14].

Here we present ChromSCape (Fig. 1), a user-friendly, step-by-step and customizable Shiny/R application to analyze all types of sparse single-cell epigenomic datasets, distributed as a Bioconductor package. The user can interactively identify sub-populations with common epigenomes within heterogeneous samples, find differentially enriched regions between subpopulations and interpret epigenomes by linking regions to associated genes and pathways. The pipeline starts from aligned sequences or count tables, and is designed for high-throughput single-cell datasets with samples containing as low as 100 cells with a minimum of 1000 reads per cell up to 25,000 cells on a standard laptop. ChromSCape accepts multiple samples to allow comparisons of cell populations between and within samples. It can determine cell identities from single-cell histone modification profiles, whatever the technology, as well as scATAC-seq datasets. We showcase the use of ChromSCape by deconvolving chromatin landscapes within the tumor micro-environment; we identify distinct H3K27me3 landscapes associated with cell identity and breast tumor subtype.

## Results

**ChromSCape identifies cell identities from scChIP-seq data.** To test the efficiency of ChromSCape (Fig. 1) in identifying cell sub-populations based on their epigenome (H3K27me3), we generated an in-silico dataset with known ground truth, mixing 4 different human cell types: Jurkat B cells, Ramos T cells, MDA-MB-468 breast cancer cells and HBCx-22 tumor cells derived from a luminal breast tumor PDX model[1]. Interestingly, Jurkat and Ramos cells were processed within the same microfluidics experiment, preventing the existence of any batch effect between them (see Grosselin et al.[1]). We compared ChromSCape to methods specifically designed for single-cell epigenomic datasets (scATAC-seq) for their ability to identify cell identities. Based on a recent scATAC-seq benchmark[9], we selected the top-performing methods, namely *Cusanovich2018*[12], SnapATAC[10], and CisTopic[11]. We also benchmarked EpiScanpy[15], a recent analysis pipeline for various single-cell epigenomic data (scATAC-seq, scDNA methylation, …) developed in Python. We applied hierarchical clustering on the reduced feature space obtained by each method and used an ARI metric to evaluate their ability to identify cell phenotypes. ChromSCape with default parameters manages to separate almost perfectly the 4 cell types, with an ARI of 0.998 (Fig. 2a), as *Cusanovich2018* and CisTopic (both an ARI of 0.996, Fig. 2b), followed closely by EpiScanpy (ARI of 0.940, Fig. 2b). ChromSCape, EpiScanpy, and SnapATAC were all run on 50 kbp bins, but SnapATAC had noisier clusters and a slightly poorer ARI (0.822).

We also compared the agility of ChromSCape to manipulate and interpret scChIP-seq datasets to two applications with graphic interface, using the same reference dataset (Fig. 2c), with either default settings or optimizing input and settings. scOrange is a stand-alone platform allowing researchers to create workflows to analyze single-cell datasets, offering a wide variety of analytical modules. While for scRNA-seq many workflows have been developed and are ready-to-use, in the case of epigenomic datasets, users need to have prior computational knowledge to organize a proper workflow. We managed to group cells

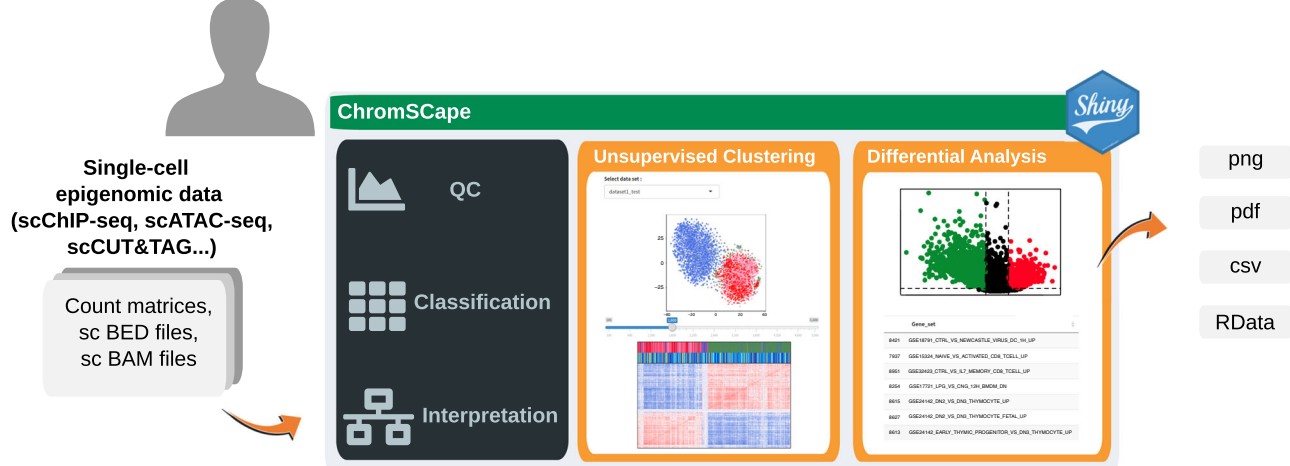

**Fig. 1 Representation of ChromSCape workflow.** Users upload single-cell epigenomic data formatted as count matrices, single-cell BAM or single-cell BED files to start the analysis. The application includes Quality Control (QC), Classification and Interpretation tools. The user can save plots and tables in png, pdf, or csv formats, and R analysis objects in RData format.

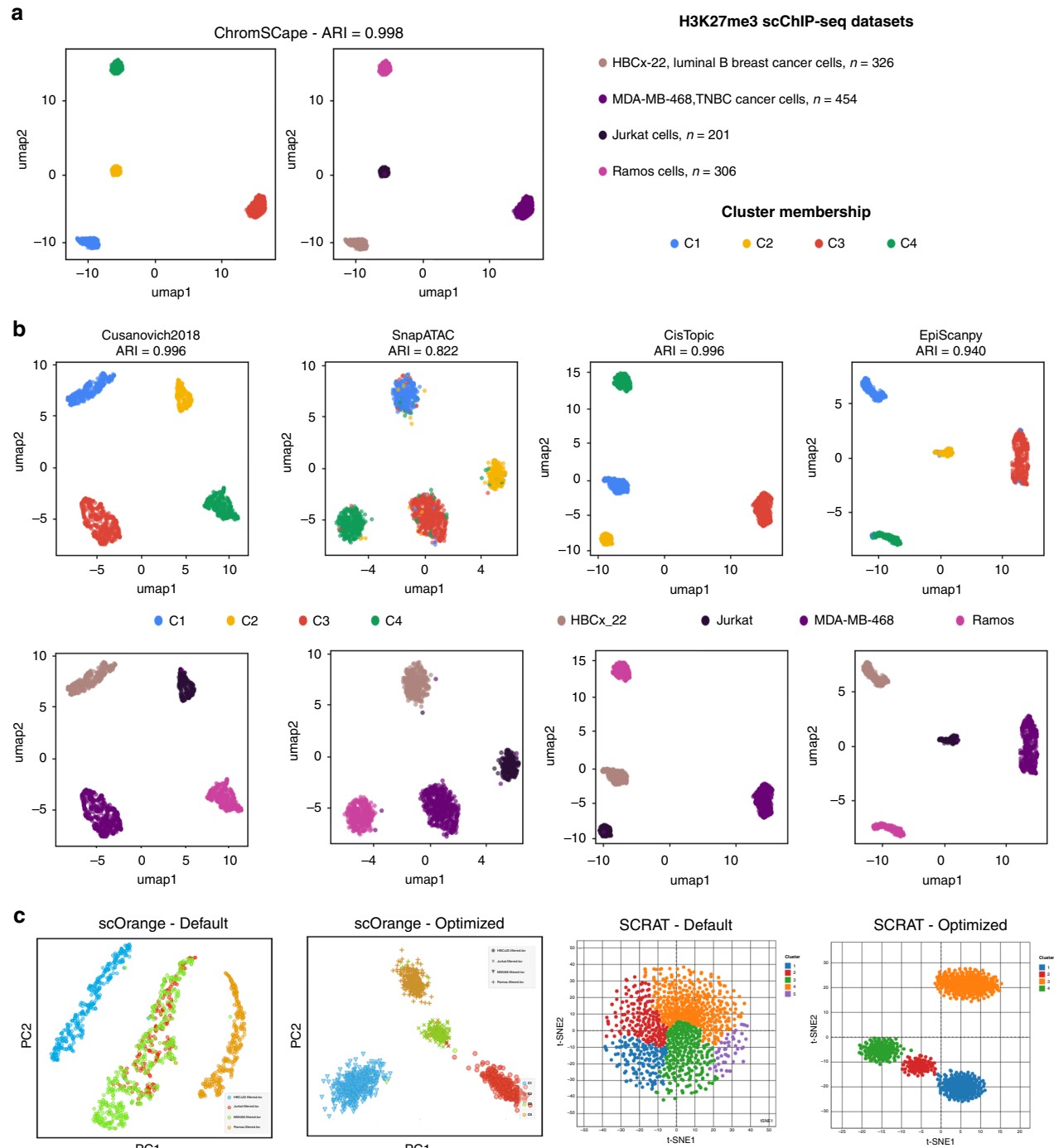

**Fig. 2 Benchmarking single-cell epigenomic tools with an in-silico mix of H3K27me3 scChIP-seq.** The mix is composed of human cells from an untreated PDX (HBCx-22), human T cells (Jurkat), and B cells (Ramos) taken from[1] and from a TNBC cell line (MDA-MB-468). (**a**) UMAP plots obtained with ChromSCape colored according to cluster and sample of origin. Adjusted Random Indexes (ARI) are indicated above the plot. (**b**) UMAP plots colored according to cluster and sample of origin with other single-cell epigenomic analysis methods: *Cusanovich2018*, SnapATAC, CisTopic, and EpiScanpy. Adjusted Random Indexes (ARI) are indicated above the plots. (**c**) Snapshots from scOrange and SCRAT applications. PCA and t-SNE representations from scOrange and SCRAT respectively, using default parameters or after manually optimizing parameters.

according to sample of origin only with an optimized workflow (Fig. 2c, default vs optimized). SCRAT is a Shiny/R package presented as a user interface to analyze single-cell epigenomic data. The default option for SCRAT is to count reads within 'ENCODE Clusters' corresponding to co-regulatory open chromatin regions obtained from DNAse-seq datasets, not adapted for the analysis of repressive histone marks like H3K27me3. In order to use SCRAT for our reference scChIP-seq datasets, we had to pre-compute counts on pre-defined peaks called on the 'pseudo-bulk' (see Methods and 'Optimized' panel), limiting the usability of SCRAT. In addition, in contrast to ChromSCape, both applications do not propose functionalities to associate genomic regions to gene annotation, limiting the biological interpretation of the results obtained with differential analysis.

Like single-cell transcriptomics approaches, single-cell epigenomic technologies can be influenced by various batch effects,

e.g., library preparation, batch of hydrogel beads or efficacy of immuno-precipitation or cleavage. To overcome this, we have implemented in ChromSCape a module for batch correction based on the fastMNN method[16]. To test this functionality, we used three datasets: two from our previous study (HBCx-95 and HBCx95-CapaR, collected with batch 1 beads) and a new dataset, HBCx95—batch 2, which is a biological replicate of HBCx-95, i.e a PDX tumor from the same PDX model but a different mouse, processed with a new batch of beads. Using ChromSCape, we analyzed together the three H3K27me3 tumor samples. As shown in Supplementary Fig. 2a strong batch effect separates the two biological replicates before batch correction (left panel). After applying batch correction (right panel), cells from the two biological replicates of untreated HBCx-95 tumors successfully mix, but not with cells from the resistant tumor, suggesting that the module corrects for batch effect without overcorrecting biological differences.

We also evaluated the usability of ChromSCape for other types of single-cell histone modification data obtained by other technologies than scChIP-seq. We analyzed two public datasets of scCUT&Tag and scChIC-seq targeting H3K27me3 and H3K4me3 marks respectively. ChromSCape facilitates the analysis of such public dataset as the user can directly upload the GEO single-cell BED files into the application. We recommend here for H3K27me3 mark—accumulating in broad peaks—to aggregate the signal into 50kbp bins, and for H3K4me3 mark—accumulating in sharp peaks—to count within 5kbp bins or around gene TSS (±2500bp). As shown in Supplementary Fig. 1a, for the scCUT&Tag dataset the two K562 replicates showed no batch effect and were clustered together separately from H1 cells (ARI = 0.976). For the scChIC-seq dataset (Supplementary Fig. 1b), 7 clusters are clearly observable on the UMAP representation, as was found by the authors in their study[4].

**ChromSCape classifies cells from scATAC-seq data.** In order to assess the capacity of ChromSCape to analyze all types of single-cell epigenomic data, we re-analyzed a scATAC-seq dataset (GSE99172) containing 8 cell lines and 4 patient-derived cells. This dataset was partly produced and analyzed in a study using chromVar, a dedicated scATAC-seq analytical tool[17]; we used the same color code as in the original study. This dataset contains various biological samples as well as technical replicates for two cell lines, K562 & GM12878, for which there are 6 and 4 technical replicates respectively. Due to a relatively low number of cells per sample ($n = 96$ per sample), we set the read count threshold to 1000 for cells to be included in the analysis. We measured the ability of ChromSCape to classify cells according to the cell type of origin using assignment scores for each sample $X$ and each cluster $Y$ (*number of cells from sample X assigned to cluster Y/total number of cells of sample X*).

In the unsupervised analysis, we identified the optimal number of clusters to be $k = 5$ according to the relative change in area under the CDF curve (Fig. 3a and Supplementary Fig. 3a). Cells from different technical replicates of K562 and GM12878 all grouped together in clusters 5 and 2 with assignment scores of 99.7% and 98.3%, respectively (Fig. 3b). Analyzing all samples together, ChromSCape could robustly identify—i.e., as stable separated clusters by consensus clustering—TF1, K562, and GM12878 cells, affecting on average 99.4% of cells to correct cluster. Cluster C1 grouped together AML, Mono, and LMPP samples with HL60 cell line, which is also originally derived from leukocytes of a patient with an AML cancer.

In order to get more insight into cell identities within cluster C1, we re-analyzed cells from C1 with ChromSCape. In contrast to the first round of analysis, ChromSCape was able to distinguish cell identities within samples, and detect individual clusters of cells, with high assignment scores for the two normal samples (LMPP & Monocytes) and HL-60 (average assignment score 98.3%, Fig. 3c, d). Additionally, AML blasts from patient SU070 show a larger proportion of monocytes than patient SU353 (Fig. 3d, $p$-value = 0.0025, Fisher's exact test, respectively 85.0% and 25.7% of SU070 and SU353 blasts cells cluster with monocytes), as previously described for these cells in[18]. ChromSCape identifies distinct populations within the normal immune cell environment based on their chromatin accessibility. Within AML patient samples, ChromSCape matches each cancer cell to the closest resembling cell in a healthy population.

**ChromSCape deconvolves epigenomes of the tumor micro-environment.** To further showcase the use of ChromSCape, we interrogated the heterogeneity of chromatin states within the tumor micro-environment of two breast tumor subtypes: luminal and triple-negative (TNBC) breast tumors. The tumor micro-environment is a key player in tumor evolution processes, and can vary between tumor types and with the response to cancer therapy. Here our goal was to compare H3K27me3 landscapes of cells from the tumor micro-environment of luminal and TNBC subtypes, resistant or not to cancer treatment. The HBCx-22 and HBCx-22-TamR datasets correspond to mouse cells from a pair of luminal ER$^+$ breast PDXs[1]: HBCx-22, responsive to Tamoxifen and HBCx-22-TamR, resistant to Tamoxifen. The HBCx-95 and HBCx-95-CapaR correspond to triple-negative breast cancer (TNBC) tumor model of acquired resistance to chemotherapy[1]. We analyzed together these four H3K27me3 mouse scChIP-seq datasets, two of which had not been analyzed in our previous study[1]. Using ChromSCape, we propose a comprehensive view of cell populations based on their chromatin profiles, and show the identification of tumor-type and treatment-specific cell populations and respective chromatin features. All plots in Fig. 4 were automatically generated by the application and are downloadable from the interface. In the quality filtering step, a threshold of 2000 reads per cell was set due to a relatively high initial number of cells ($n = 5516$ cells).

After the dimensionality reduction step (Fig. 4a, b), we applied our consensus clustering approach on the filtered dataset with $k = 2$ to $k = 10$ clusters. We chose to partition the data into $k = 4$ clusters based on the knee method, as a plateau in the relative change in area under the CDF curve was observed between $k = 4$ and $k = 5$ clusters (Fig. 4c, d & Supplementary Fig. 4a, b). Consensus score matrix in Fig. 4d shows that most of the cells were stably assigned to four chromatin-based populations (mean consensus score for selected clusters of 0.91 which is significantly higher than mean consensus score for other clusters, 0.17, $p$-value = 2.2e-16, two-sided Student's $t$-test). Assignment of cells to cluster C2 and C4 is significantly less stable than C1 and C3 ($p$-value < 2.2e-16, Student's two-sided $t$-test, mean consensus scores are respectively 0.84 and 0.90 for C1-C3 and 0.70 and 0.71 for C2-C4, see Supplementary Fig. 3a), suggesting that cells from C2 and C4 might share H3K27me3 features, whereas cells from C1 and C3 have distinct H3K27me3 landscapes. Clusters C1, C2, and C4 contain cells from all four samples, with a significantly higher proportion of HBCx-22-TamR for C1 ($p$-value = 3e-05, Pearson's Chi-squared test) (Fig. 4e). On the other hand, cluster C3 is almost exclusively composed of cells from model HBCx-95 (Fig. 4c, e), revealing a stromal cell population specific to the triple-negative breast cancer model (HBCx-95).

To further identify the specific features of each chromatin-based population, we proceeded to peak calling, differential analysis, and gene set enrichment analysis using default

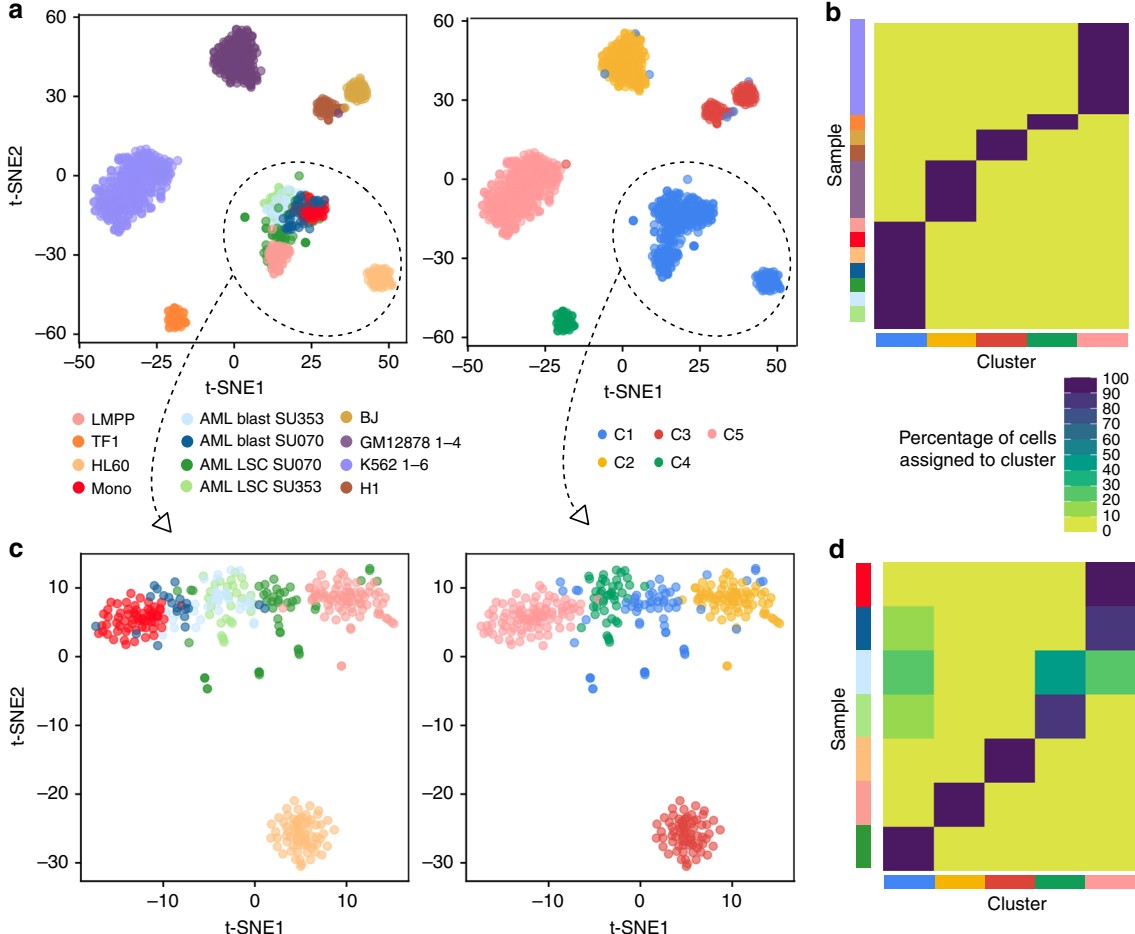

**Fig. 3 ChromSCape identifies immune cell populations from scATAC-seq datasets. (a)** t-SNE representations after correlation filtering ($n = 1309$ cells), points are colored according to sample of origin (left) or ChromSCape-determined clusters ($k = 5$) (right). The GM12878 and K562 samples contained respectively 4 and 6 replicates. (**b**) Assignment scores for each sample/cluster pair for the analysis with all samples. (**c, d**) As in (**a**) and (**b**) for the analysis with only AML, LSC, monocyte, LMPP & HL60 cells ($n = 347$ cells).

parameters (see Methods). As H3K27me3 is a repressive histone mark, we focused our analysis on loci depleted in H3K27me3, where transcription of genes can occur, in cells from each cluster versus all other cells. The differential analysis identified respectively 189, 210, 83, and 9 significantly depleted regions for clusters C1 to C4 (Fig. 4g, logFC <1, adjusted $p$-value < 0.01). We found loci devoid of H3K27me3 specific to cluster C2, enriched for genes involved in apical junction such as *Bcar1* (Fig. 4f) and *Ptk2*, which are characteristic of genes expressed in fibroblasts. We found a depletion of H3K27me3 specific to cluster C3 over the genes *Nrros* (Fig. 4f) and *Il10ra*, two genes characteristic of immune expression programs. The depletion of H3K27me3 over the transcription start site of *Rap1gap2*, a gene expressed in endothelial cells, was a key feature of cluster C4 (Fig. 4f). For cluster C1 and C2, we found a depletion of H3K27me3 over *Eln*, a gene expressed in fibroblasts.

Gene set enrichment analysis for genes located in regions depleted of H3K27me3 enrichment only revealed very few enriched gene lists, mostly for cluster C2 ($q$-value < 0.1, Fig. 4h, multiple gene sets related to stem and cancer cells) and one list for C1 ("LPS_VS_CONTROL_MONOCYTE_UP"). Linking H3K27me3 enrichment to transcription is indeed indirect, we envisage such enrichment analysis more appropriate for H3K4me3 scChIP-seq in which enriched regions are directly associated with gene transcription.

Overall, these results are consistent with our previous analysis of HBCx-95 scRNA-seq datasets where subpopulations were differentially expressing markers of fibroblasts, endothelial, and macrophage cells[1]. This analysis comprising the HBCx-22 dataset allowed us to identify the H3K27me3 signature of potential endothelial cells (cluster C4). These cells are present in each model, but might not have been previously detected in the previous scChIP-seq analysis due to low cell representation. In addition, the H3K27me3 signature of potential immune cells is restricted to cells from the TNBC model (cluster C3), suggesting that these immune cells are absent from the luminal tumor.

## Discussion

ChromSCape is a Shiny/R application designed for both biologists and bioinformaticians to analyze complex chromatin profiling datasets such as scChIP-seq datasets. The comprehensive application is quick to take over plus the direct visualization of cells clusters combined to configurable parameters and incremental saving of intermediary R objects eases bench-marking of parameters. We show that ChromSCape performs as well or better than state of the art single-cell epigenomic analytic tools to identify cell identities from an in-silico mix of H3K27me3 scChIP-seq datasets. It also manages to identify sub-populations within a complex scATAC-seq benchmarking dataset, showing its wide range of application for epigenomic analysis. In addition,

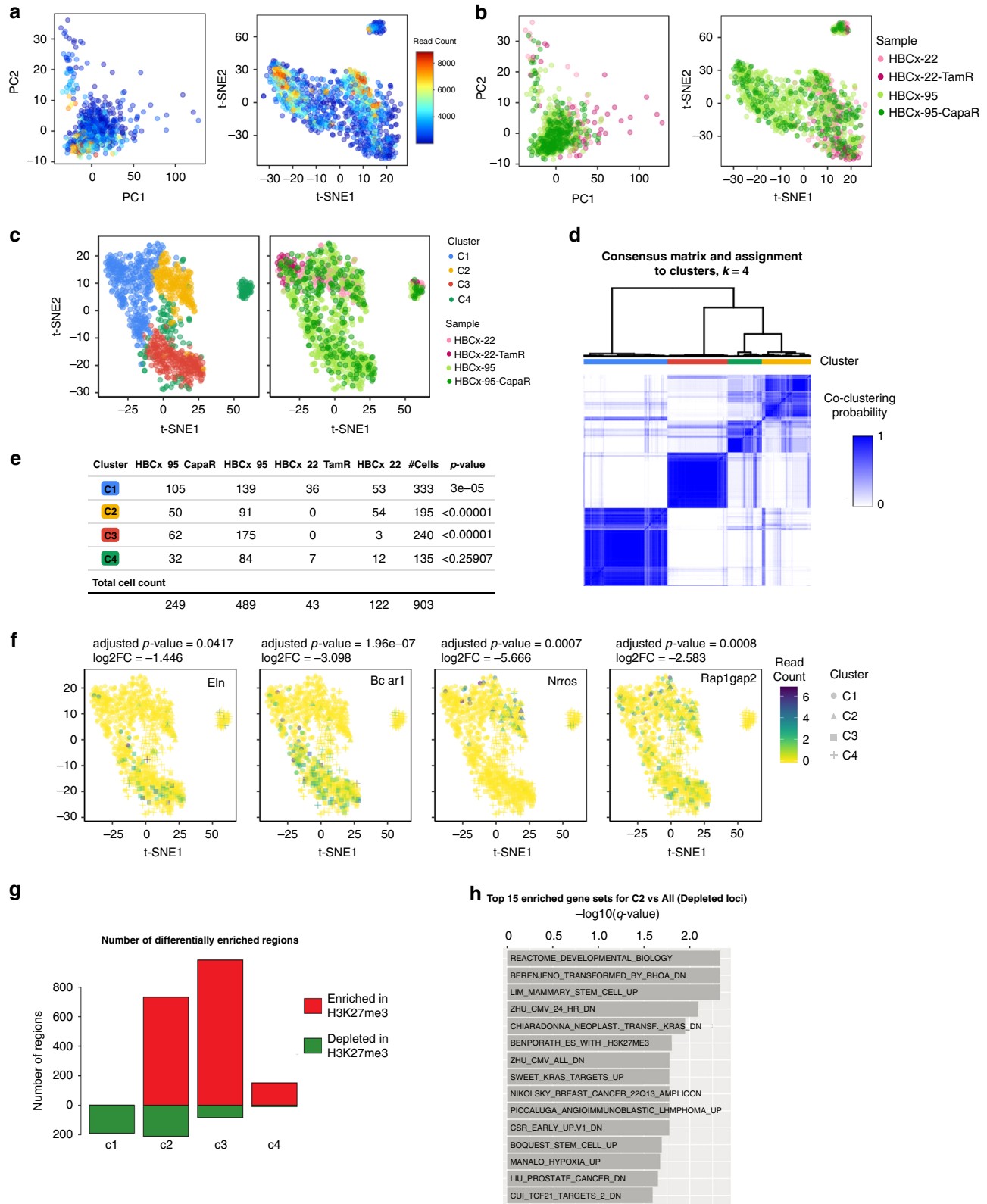

using ChromSCape to study the epigenome of mouse stromal cells in breast tumors, we can identify the various epigenomes within the tumor micro-environment. Overall, we see ChromSCape as a useful tool to probe heterogeneity and dynamics of chromatin profiles in various biological settings, not only in cancer development but also in cell development and cellular differentiation.

## Methods

**Implementation.** ChromSCape is a Bioconductor R package developed in Shiny/R. It uses various Shiny related packages (shinyjs, shinydashboard, shinyDirector-yInput) for the user interface. The application takes advantage of public R libraries for data vizualisation (RcolorBrewer, colorRamps, Rtsne, umap, colourpicker, kableExtra, knitr, viridis, ggplot2, gplots, png, grid, gridExtra, DT) as well as for data manipulation (Matrix, dplyr, tidyr, stringr, irlba, rlist, qualV, stringdistr). ChromSCape uses Bioconductor packages (i) for the manipulation of single-cell

**Fig. 4 ChromSCape deconvolves epigenomic landscapes within the tumor micro-environment.** Cells belonging to samples HBCx-22, HBCx-22-TamR, HBCx-95, HBCx-95-CapaR PDX[1]. PCA and t-SNE plots are colored according to the amount of uniquely mapped reads per cell (**a**) and to sample of origin (**b**). (**c**) t-SNE representations after correlation filtering (n = 903 cells), colored by cluster or sample of origin. (**d**) Hierarchical clustering and corresponding heatmap of cell-to-cell consensus clustering scores cells portioning the dataset into k = 4 clusters. Consensus score ranges from 0 (white: never clustered together) to 1 (dark blue: always clustered together). Cluster membership is color-coded above the heatmap. (**e**) Table of cluster memberships. P-value column results from Pearson's Chi-squared goodness of fit test without correction, checking if the observed distribution of samples in each cluster differs from a random distribution. Source data are provided as a Source Data file. (**f**) t-SNE representation of scChIP-seq datasets, points are colored according to H3K27me3 enrichment signals in each cell for genes located within depleted regions in C1 to C4, respectively *Eln*, *Bcar1*, *Nrros*, and *Rap1gap2*. The adjusted p-values and log2FC of the associated regions are indicated above each plot. (**g**) Barplot of differentially enriched regions identified by Wilcoxon signed-rank test. Genomic regions were considered enriched (red) or depleted (green) in H3K27me3 if the adjusted p-values were lower than 0.01 and the absolute fold change greater than 1. (**h**) Barplot displaying the -log10 of adjusted p-values from pathway analysis for cells of cluster C2 compared to all other cells in depleted loci. Only the top 15 significant gene sets, ranked by adjusted p-values, are indicated.

data with SingleCellExperiment, scater[19], scran[20], (ii) for the manipulation of genomic regions with IRanges and GenomicRanges[21], (iii) for the manipulation of genomic files with Rsamtools and BiocParallel, (iv) for the correction of batch effects with batchelor[16] and (v) to determine the optimal number of clusters with ConsensusClusterPlus[22]. In addition, ChromSCape makes use of custom R functions which serve for both manipulation and visualization of datasets. Brief command lines enable users without any bioinformatics skills to install all R dependencies and run the application in a web browser.

**Demonstration application.** A demonstration of ChromSCape is freely available at https://vallotlab.shinyapps.io/ChromSCape/.

**Input datasets, quality control, and pre-processing.** Input files for ChromSCape are either one or multiple count matrices with genomic regions in rows and cells in columns or single-cell BAM or BED files. In this case, a directory containing single-cell BAM or BED files must be specified and ChromSCape creates the count matrix by aggregating the signal into successive genomic bins, peaks (BED file must be provided by user) or into regions around genes Transcription Start Sites (TSS). For H3K27me3 scChIP-seq datasets, with a distribution in broad peaks, we recommend using bins of 50kbp, while for H3K4me3 scChIP-seq or scATAC-seq datasets we recommend using smaller bins (e.g., 5 kb), known peaks or regions around TSS. The "condition" or "label" of each cell is then heuristically determined using file names and the number of conditions specified by the user. Guidelines and links toward datasets are given in the user guide (https://vallotlab.github.io/ChromSCape/ChromSCape_guide.html).

In order to efficiently remove outlier cells from the analysis, e.g., cells with excessively high or low coverage, the user sets a threshold on a minimum read count per cell and the upper percentile of cells to remove. The latter could correspond to doublets, e.g., two cells in one droplet, while lowly covered cells are not informative enough or may correspond to barcodes ligated to contaminant DNA or to library artifacts. Regions not supported by a minimum user-defined percentage of cells that have a coverage greater than 1000 reads are filtered out. Defaults parameters were chosen based on the analysis of multiple scChIP-seq datasets from our previous study[1]: a minimum coverage of 1600 unique reads per cell, filtering out the cells with the top 5% coverage and keeping regions detected in at least 1% of cells. Post quality control filtering, the matrices are normalized by total read count and region size. At this step, the user can provide a list of genomic regions, in BED format, to exclude from the subsequent analysis, in cases of known copy number variation regions between cells for example.

To reduce the dimensions of the normalized matrix for further analysis, principle component analysis (PCA) is applied to the matrix, with centering, and the 50 first PCs are kept for further analysis. The user can visualize scChIP-seq data after quality control in the PCs dimensional space. The t-distributed stochastic neighbor embedding (t-SNE) algorithm[23] and UMAP[24] is applied on the PCA to visualize the data in two dimensions. The PCA and t-SNE plots are a convenient way to check if cells form clusters in a way that was expected before any clustering method is applied. For instance, the user should verify whether the QC filtering steps and normalization procedures were efficient by checking the distribution of cells in PC1 and PC2 space. Cells should group independently of normalized coverage. In our hands, for our scChIP-seq H3K27me3 datasets, minimum coverage of 1,600 unique reads per cell was required to separate cells independently of coverage post normalization[1]. A batch correction option using mutual nearest neighbors "FastMNN" function from "batchelor" package[16] is implemented to remove any known batch effect in the reduced feature space.

**Hierarchical clustering, filtering, and consensus clustering.** Using the first 50 first PCs of computed PCA as input, hierarchical clustering is performed, taking 1-Pearson's correlation score as the distance metric. To improve the stability of our clustering approaches and to remove from the analysis isolated cells that do not belong to any subgroup, cells displaying a Pearson's pairwise correlation score below a threshold t with at least p% of cells are filtered out (p is set at 1% by default). The

correlation threshold t is calculated as a user-defined percentile of Pearson's pairwise correlation scores for a randomized dataset (percentile is recommended to be set as the 99th percentile). Correlation heatmaps before and after correlation filtering and the number of remaining cells are displayed to inform users on the filtering process.

ChromSCape uses Bioconductor ConsensusClusterPlus package[22] to determine what is the appropriate k-partition of the filtered dataset into k clusters. To do so, it evaluates the stability of the clusters and computes item consensus score for each cell for each possible partition from k = 2 to 10. For each k, consensus partitions of the dataset are done on the basis of 1000 resampling iterations (80% of cells sampled at each iteration) of hierarchical clustering, with Pearson's dissimilarity as the distance metric and Ward's method for linkage analysis. The optimal number of clusters is then chosen by the user; one option is to maximize intra-cluster correlation scores based on the graphics displayed on the "Consensus Clustering" tab after processing. Clustering memberships can be visualized in two dimensions with the t-SNE or UMAP plot.

**Peak calling for genomic region annotation.** This step of the analysis is optional, but recommended in order to refine the peak annotation prior to enrichment analysis. To be able to run this module, MACS2 is required[25]. The user needs to input BAM files for the samples (one separate BAM file per sample), with each read being labeled with the barcode ID. ChromSCape merges all files and splits them again according to the previously determined clusters of cells (one separate BAM file per cluster). Customizable significance threshold for peak detection and merging distance for peaks (defaults to p-value = 0.05 and peak merge distance to 5,000) allows to identify peaks in close proximity (<1000 bp) to a gene transcription start site (TSS); these genes will be later used as input for the enrichment analysis. For the annotation, ChromSCape uses the reference human transcriptome Gencode_hg38_v26, limited to protein-coding, antisense, and lncRNA genes.

**Differential analysis and pathway enrichment analysis.** To identify differentially enriched regions across single-cells for a given cluster, ChromSCape can perform (i) a non-parametric two-sided Wilcoxon rank-sum test comparing normalized counts from individual cells from one cluster versus all other cells, or cluster of choice, or (ii) a parametric test comparing raw counts from individual cells, using edgeR[26], based on the assumption that the data follows a negative-binomial distribution. We test for the null hypothesis that the distribution of normalized counts from the two compared groups has the same median, with a confidence interval 0.95. The calculated p-values are then corrected by the Benjamini–Hochberg procedure[27]. The user can set a log2 fold-change threshold and corrected p-value threshold for regions to be considered as significantly differentially enriched (default settings are a p-value and log2 fold-change thresholds respectively of 0.01 and 1). If users have specified batches, the differential analysis is done using the "pairwiseWilcox" function from the scran package[20], setting the batch of origin as a "blocking level" for each cell.

For the top 100 most significant differential regions, single-cell H3K27me3 enrichment levels can be visualized overlaying H3K27me3 counts for each cell at selected genes onto a t-SNE plot. Using the refined annotation of peaks done in the previous step, the final step is to look for enriched gene sets of the MSigDB v5 database[28] within differentially enriched regions (either enriched or depleted regions in the studied histone mark). We apply hypergeometric tests to identify gene sets from the MSigDB v5 database over-represented within differentially enriched regions, correcting for multiple testing with the Benjamini–Hochberg procedure. Users can then visualize the most significantly enriched or depleted gene sets corresponding to the epigenetic signatures of each cluster and download gene sets enrichment tables.

**Datasets.** H3K27me3 scChIP-seq human in-silico mix of 4 cell types: The samples correspond to n = 326 human tumor cells from untreated PDX (HBCx-22), n = 201 human T cells (Jurkat) and n = 306 B cells (Ramos) taken from[1] and n = 454 cells from the MDA-MB-468 triple-negative breast cancer cell line (HBCx-22, Jurkat and Ramos data are from "GSE117309"), MDA-MB-468 is available at "GSE152502").

H3K4me3 scChIC-seq human white blood cells dataset[4]: $n = 285$ white blood cells from a human male donor were downloaded as gzipped single-cell BED files from "GSE105012", inputted directly into ChromSCape and aggregated into 50kbp bins (default).

H3K27me3 scCUT&Tag human H1 and K562 cells[5]: A replicate of K562 cell line comprising of $n = 908$ cells from "GSE124680", another replicate of $n = 479$ K562 cells and $n = 486$ H1 cells from "GSE124690" were downloaded as gzipped single-cell BED files, inputted directly into ChromSCape and aggregated around gene TSS (±2500 bp).

H3K27me3 scChIP-seq human datasets: The samples correspond to human cells from patient-derived xenograft (PDX) originating from two different human donors[1]. For this study, we added a new scChIP-seq dataset, corresponding to a biological replicate of HBCx-95 ("GSE152502"), processed with a novel batch of hydrogel beads.

scATAC-seq datasets: The scATAC-seq dataset is composed of two cell types derived from two acute myeloid leukemia (AML) (patients SU070 and SU353 blastocytes (blast) and leukemic stem cells (LSC) from[18] as well as multiple cell lines: GM12878 (4 replicates), TF1, BJ, H1, HL60, K562 (3 replicates) from[29], K562 (3 replicates) from[17]; monocytes (Mono) and lymphoid primed multipotent progenitor (LMPP) from[18]. The count matrix of reads in peaks was downloaded from GEO accession number "GSE99172", split into distinct matrices for each sample and formatted to be accepted as input by ChromSCape.

H3K27me3 scChIP-seq mouse datasets: The samples correspond to mouse cells from patient-derived xenograft (PDX) originating from two different human donors[1]. Raw FASTQ reads were processed using the latest version of our scChIP-seq data engineering pipeline (see above) to produce 50 kbp binned count matrices given as input to ChromSCape (matrices available at https://figshare.com/projects/Single-Cell_ChIP-seq_of_Mouse_Stromal_Cells_in_PDX_tumour_models_of_resistance/66419).

**Cell line**. MDA-MB-468 cells, bought at ATCC (HTB-132™), were cultured in DMEM 1640 (Gibco-BRL) and supplemented with 10% heat-inactivated fetal calf serum. Cell numbers, as judged by Trypan Blue exclusion test, were determined by counting cells using a Countess automated cell counter (Invitrogen). Cells were cultured at 37 °C in a humidified 5% $CO_2$ atmosphere. The cell line was myco-plasma negative. The MDA-MB-468 cells were trypsinized (Trypsin, Gibco-BRL). Prior to single-cell ChIP-seq, cells were then re-suspended in PBS/0.04% BSA (ThermoFisher Scientific, # AM2616).

**Patient-derived xenograft (PDX)**. Female Swiss nude mice were purchased from Charles River Laboratories and were maintained under specific pathogen-free conditions. Their care and housing were in accordance with institutional guidelines and the rules of the French Ethics Committee (project authorization no. 02163.02). A PDX from a residual triple-negative breast cancer post neo-adjuvant che-motherapy (HBCx-95) was previously established at Institut Curie with informed consent from the patient[1]. Prior to single-cell ChIP-seq, PDX was digested at 37 °C for 2 h with a cocktail of Collagenase I (Roche, # 11088793001) and Hyaluronidase (Sigma, # H3506). Cells were then individualized at 37 °C using a cocktail of 0.25% trypsin/Versen (ThermoFisher Scientific, #15040-033), Dispase II (Sigma, #D4693), and Dnase I (Roche, # 11284932001). Red Blood Cell lysis buffer (ThermoFisher Scientific, # 00-4333-57) was then added to degrade red blood cells. In order to increase the viability of the cell suspension, dead cells were removed using the Dead Cell Removal kit (Miltenyi Biotec). Cells were re-suspended in PBS/0.04% BSA (ThermoFisher Scientific, # AM2616).

**Single-cell ChIP-seq**. The protocol for scChIP-seq was rigorously the same as in Grosselin et al.[1], and can be resumed by the main following steps. Cells were first compartmentalized into droplets containing Mnase in a microfluidics chip, then fused with barcoded hydrogel beads. After fusion of cell-containing droplets and bead-containing droplets, Fast-link DNA ligase [Lucigen, # LK0750H] was used to ligate segmented DNA to barcodes. Droplets were pooled and used for chromatin immuno-precipitation with 2.5 µl of anti-H3K27me3 antibody ([Cell Signaling Technology, # 9733]). After treatment with RNAse A (ThermoFisher Scientific, #EN0531) and Proteinase K (ThermoFisher Scientific, # EO0491), barcoded-nucleosomes were then amplified by in-vitro transcription using the T7 MegaScript kit (ThermoFisher Sci-entific, # AM1334) and reverse-transcribed. After RNA digestion, DNA was amplified by PCR. The final product was size-selected by gel electrophoresis. Single-cell ChIP-seq libraries were finally sequenced on an Illumina NextSeq 500 MidOutput 150 cycles.

**Demultiplexing and alignment of H3K27me3 scChIP-seq datasets**. Raw FASTQ reads were processed using the latest version of our scChIP-seq data engineering pipeline that allowed a more precise removal of PCR and RT dupli-cates (code available at https://github.com/vallotlab/scChIPseq_DataEngineering) to produce 50 kbp binned count matrices given as input to ChromSCape (matrices available at https://figshare.com/projects/Single-Cell_ChIP-seq_of_Mouse_Stromal_Cells_in_PDX_tumour_models_of_resistance/66419). Rapidly, the first 56 bp of the Read2 were separated into three indexes and aligned using

bowtie2 separately against the reference of three pools of 96 16-bp long indexes. Reads containing all three recognizable indexes (a full cell-barcode) were kept, the genomic part of Read2 and Read1 were aligned in paired-end mode using STAR v2.7.0. For each barcode, aligned reads were deduplicated by removing successively: (i) PCR duplicates, identified if #Read1 + #Read 2 mapped at the same position, (ii) RT duplicates, identified if #Read 1 mapped at the same position, and (iii) window duplicates: all the reads falling in the same 50 bp window were stacked into one as reads possibly originating from the same nucleosome. Reads were binned in non-overlapping 50 kb bins spanning the genome to generate a n x m coverage matrix with n barcodes and m genomic bins used in downstream analysis.

**Benchmark of tools for scChIP-seq data analysis**. Three methods dedicated to the analysis of scATAC-seq with the best performance according to Chen et al., 2019[9], were tested on a mixture of H3K27me3 scChIP-seq datasets (see Datasets below), namely "SnapATAC", "CisTopic", and "Cusanovich2018". The scripts were taken from the GitHub repository of the benchmark paper (https://github.com/pinellolab/scATAC-benchmarking). For "CisTopic" and "Cusanovich2018", peaks were called using MACS2 with options "-nomodel -extsize 300 -keep-dup all -broad". Peaks closer than 5000 bp were merged together using BEDTools. For "SnapATAC", 50kbp bins were counted from BAM files using "SnapTools". In addition, we also tested a recent method for single-cell epigenomic analysis, "EpiScanpy", following the basic steps described in the tutorial for scATAC-seq (https://github.com/colomemaria/episcanpy/blob/master/docs/tutorials/Tutorial_Hackathon_Buenrostro_2.html) with the same 50 kbp matrices used for ChromSCape. We extracted from each method the matrix of reduced feature space, and used hierarchical clustering with Pearson's dissimilarity as the distance metric and Ward's method for linkage. The adjusted Rand's index (ARI), a widely used measure to quantify clustering accuracy, was calculated for each method using R package "mclust"[30], taking samples of origin as "true" clusters.

In addition, two softwares with graphic interface, dedicated to the analysis of single-cell data, "scOrange"[13] and "SCRAT"[14], were also tested on the same set of cells both with "default" parameters and manually "optimized" parameters. default' for scOrange corresponds to using the template called "Loading data from 10× protocols", a workflow meant for analyzing scRNA-seq of single cells, replacing the input by our matrices of selected cells in 50kbp bins. The "optimized" workflow is available at www.github.com/vallotlab/ChromSCape_benchmarking and can be opened with the "scOrange" software. For 'SCRAT', we found that the "optimized" counting method corresponded to counting signal within peaks called on the "pseudo-bulk" (see above).

In order to be able to compare the distinct methods, ChromSCape was first to run on the raw count matrices and a set of 1287 cells passing the quality control thresholds were selected to be used as input for all methods. As the number of cells in each sample was unbalanced (e.g., the raw MDA-MB-468 containing $n = 3,382$ cells while others have a maximum of $n = 456$ cells), 500 cells from MDA-MB-468 were randomly sub-sampled using ChromSCape "Perform Subsampling" option. We removed from the analysis the segments corresponding to known amplifications and homozygous loss of DNA of the Triple Negative Breast Cancer cell line MDA-MB-468, corresponding to a total of 77Mbp, previously found by analyzing the input of bulk ChIP-seq of the same cells (see Supplementary Note 2).

**Reporting Summary**. Further information on research design is available in the Nature Research Reporting Summary linked to this article.

## Data availability
In this study, we produced H3K27me3 scChIP-seq data for MDA-MB-468 sample and a new replicate of untreated HBCx-95. This sequencing data have been deposited in the National Center for Biotechnology Information Gene Expression Omnibus (GEO) and are accessible through the GEO Series accession number "GSE152502". Other datasets used (described in "Datasets" in the Methods) can be downloaded from NCBI GEO under the accession numbers "GSE117309", "GSE152502", "GSE105012", "GSE124680", "GSE124690", "GSE152502", "GSE99172". All other relevant data supporting the key findings of this study are available within the article and its Supplementary Information files or from the corresponding authors upon reasonable request. A reporting summary for this Article is available as a Supplementary Information file.

## Code availability
The package is available on Bioconductor v3.12, requiring R4.0. Source code, guidelines for installation, and use of the application are provided at https://github.com/vallotlab/ChromSCape. A docker container containing the application and it's dependecies is available on DockerHub (pacomito/chromscape:v0.0.9001), instructions on how to launch it are available on the github page. Codes for the benchmark of "SnapATAC", "CisTopic", "Cusanovich2018", and "EpiScanpy" are available at https://github.com/vallotlab/ChromSCape_benchmarking.

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

## Acknowledgments

This research project was supported by the ATIP Avenir program, by Plan Cancer and by the SiRIC-Curie program SiRIC Grants #INCa-DGOS- 4654 and #INCa-DGOS-Inserm_12554.

## Author contributions

P.P., P.K., and C.V. wrote code for ChromSCape and performed data analysis. J.M. performed scChIP-seq experiments on PDX and breast cancer cell lines. P.P., M.D., and N.S. processed raw scChIP-seq datasets into count matrices. P.P. and C.V. wrote the manuscript. C.V. conceptualized and supervised this work.

## Competing interests

The authors declare the following competing interests: licenses have been filed on some aspects of this work by Institut Curie and CNRS; contributors may receive payments related to exploitation of these licenses under their employer's rewards to inventor scheme.
