## [Peer Review File · Nature Communications]

Reviewers' comments:

Reviewer #1 (Remarks to the Author):

Prompsy, Kirchmeier et al. present ChromScape a web application for the analysis of single cell ChIP-seq or ATAC-seq datasets. The application is aimed to researcher without bioinformatic expertise. ChromScape implements common steps for the analysis and visualization of count matrices.

The webapp is well made and in principle could be very helpful, however I think it is important to address these points before publication.

Major points:

1) If the goal is to provide an end to end pipeline for people without scripting skills it is necessary to start from raw sequencing data and not from count data. These steps should be included in the current pipeline and integrated in the Shiny user interface. This can be easily accomplished by integrating the other library you have created:

https://github.com/vallotlab/scChIPseq_DataEngineering

2) Although the goal is to provide a user-friendly graphical application it is not clear what is the quality of the results produced. To this end it is important to compare the results obtained with this pipeline with recent pipelines even though they don't provide a web-based interface.

- For ChIP-seq data, please compare your results with methods also based on count matrices like EpiScanpy (<https://europepmc.org/article/ppr/ppr80155>) or Seurat (https://satijalab.org/seurat/v3.0/atacseq_integration_vignette.html).

- For ATAC-seq data in addition to EpiScanpy several methods have been proposed. A systematic comparison is presented in Chen et al 2019 (<https://genomebiology.biomedcentral.com/articles/10.1186/s13059-019-1854-5>). From their analysis it is clear that it is not necessary to compare ChromScape with all the available recent methods but only with the top performing i.e. SnapATAC, Cusanovich 2018 and CisTopic. Without this comparison it is hard to assess the quality of the proposed results.

3) "No batch correction option is implemented in this version of the application." Explain the implications of this, warn the users and suggest some alternatives to remove batch effects using available packages. Alternatively, consider including a simple method for batch correction in ChromScape, for example, the method developed by the Marioni lab (<https://www.ncbi.nlm.nih.gov/pubmed/29608177>) based on mutual nearest neighbors.

4) Please provide some information on memory requirement and running time for the datasets presented. Also, what is the maximum count matrix size a user can analyze on a personal computer with 8 or 16GB or memory?

5) The peak analysis is disconnected from the rest, since the tool takes in input count matrices, however for this step is necessary to obtain bam files and to have experience with command line tools and scripting. As mentioned before to make ChromScape truly useful to non-expert, it is necessary to start from raw sequencing data.

6) The support for ATAC-seq data is very primitive and limited. The assumption again is to start from a count matrix and some custom scripts were necessary to proceed with the analysis "The count matrix of reads in 188 peaks was downloaded from GEO accession number GSE99172, split into distinct matrices for 189 each sample and formatted to be accepted as input by ChromScape." If you want to support scATAC-seq datasets it is necessary to include the preprocessing steps to

get a matrix from raw reads.

Minor points:

1) Line 286: extra space in "Ad ditionally"

2) Windows support? In the docker version clear instructions are provided for Linux/Mac OS, however no instructions are provided for Windows. This is a missing opportunity given that Docker is also supported on Windows and a simple modification can be provided to support also this platform.

Reviewer #2 (Remarks to the Author):

Recent technological advancement enables epigenetics analysis at a single-cell level, but analysis tools dedicated for sparse single-cell epigenome datasets are currently lacking and thus desirable. In this study, Prompsy et al. developed ChromSCape, a GUI application that is designed for single-cell data and allows e.g. identification of subpopulations of cells based on epigenome states. Using ChromSCape, the authors showed the ability of their software that accepts count matrix of genomic bins (cells x bins) and automates the following data analysis steps including filtering outlier cells, lower-dimensional representation, clustering, calling peaks and differential enriched regions. They performed demonstrations on scChIP-seq data from their previous report as well as scATAC-seq data used for another scATAC-seq analytical tool chromVar, showing ChromSCape worked well on both datasets. Thus, this work has a potential to liven the blooming field of single-cell epigenomics by providing a user-friendly analysis tool. At this stage, however, the software seems immature and specifically dedicated to their own droplet-based scChIP-seq technology. This manuscript merely introduces their data analysis tips and a combo of BioConductor tools, rather than proposes novel computational/statistical ideas and/or technical advancements in the sparse and high-dimensional single-cell epigenomic data analysis. Overall, although ChromSCape could be a valuable tool for single-cell epigenome analysis, the manuscript in its current form does not necessarily need to be published on the journal Nature Communications and thus may be suitable for other more specific journals.

Specific points

1: The authors stated that "With ChromSCape (Fig. 1), we propose a user-friendly, step-by-step and customizable R/Shiny application to analyze sparse single-cell histone modifications datasets. (line 54)". Setting up ChromScape, however, appears to still require high computational skills, especially for general biologists. More importantly, installing the application does not work in my brief trial (both R environment and docker container, a point related to path resolving problem after uploading count matrix). This implies this software has not been well tested.

2: ChromScape does not provide a general guideline for a wide variety of single-cell epigenomic technologies recently developed, such as CUT&Tag, scChIL, scChIC and CoBATCH. It seems the authors merely follow droplet-based scRNA-seq data analysis (e.g., "In order to efficiently remove outlier cells from the analysis, e.g. cells with excessively high or low coverage, the user sets a threshold on a minimum read count per cell and the upper percentile of cells to remove. (line 92)"). The authors should provide a universal exploratory data analysis platform for e.g. deciding filtering threshold, optimizing bin-size, correction of unwanted variations and batch-effects. Otherwise its applicability and/or robustness on many other single-cell epigenomic datasets have not sufficiently been demonstrated.

3: (Calling differentially enriched regions) Comparing one specified cluster with all the others seems less sensitive for detecting candidates because of the large variability of out-of-cluster cells. Providing options to test all possible pairs of clusters will be useful for users.

4: (Statistical modeling for categorical variables) Why do the authors prefer non-parametric (Wilcoxon rank sum) tests over Poisson or Negative binomial modeling of count data? If recommended, the authors should compare these different statistical approaches with regard to sensitivity/specificity.

Reviewer #3 (Remarks to the Author):

Prompsy and colleagues describe in their manuscript a software (called ChromSCape) for the analysis of single-cell ChIP-seq data and illustrate the use with data from their recent publication (Grosselin et al. 2019, Nat. Genet.). Overall, I believe the chosen examples serve the illustrative purpose and I appreciate the attempt to create user-friendly software for single-cell data analysis, but I believe the solution falls completely short of providing a truly user-friendly tool in its current form. The authors also need to add a proper comparison to the state-of-the-art. If these issues can be resolved / improved upon, I would be supportive for publication in Nature Communications.

Major points -- software:

- The tool is primarily targeted at experimental biologists with limited bioinformatics skills (“[to] enable users without any bioinformatics skills [...] run the application in a web browser”). I very much doubt that the average target user will be able to install the software in its basic setup (R, bedtools, command line execution) and even less so with additional packages required for advanced analysis (SAMtools, bedtools, MACS2, for peak calling l. 144ff.). For me, the installation script asking me to downgrade my Bioconductor release, which I of course chose not to do to avoid conflicts with other projects. Setting up an R install just for ChromSCape is a big ask. Really, the authors should implement ChromSCape as a Bioconductor package which will force them to mind dependencies and versioning.

Providing a Docker image with pre-installed software requirements is a good thought in order to ease the installation burden (this is what I chose to do after the problems I described above), however, this first requires Docker to be installed and admin rights on the computer, plus being comfortable with setting up directories and changing permissions –again considerable stumbling stones for many prospective users.

All installation problems can probably be managed with additional time investment and help of computer-savvy colleagues, but a quick and easy way to test and trial the software is required for potential users to evaluate whether it’s worth to go through all that trouble.

I would therefore highly recommend setting up a public demo server with the application that allows immediate execution with a trial dataset.

- Even after installing the Docker version of the software from Docker Hub and following the instructions on the GitHub page about creating directories and running as root, I faced difficulties using it (I used Ubuntu Linux). Uploading new datasets (I used the count matrices provided on figshare) consistently resulted in disconnection from the server. I then tried to download the precompiled analysis (also provided by the authors on figshare) and tried putting it in the Data_ChomSCape folder (no detailed instructions provided what else to do with it) – I could not see it from the running application. I gave up at this point, having spent a considerable amount of time without being able to try the software.

It seems to be that these kind of issues should be resolved by the authors by some proper software testing. They should install the software following their own instruction on a "clean" computer – i.e. one that is not already set up with all their commonly used tools and configuration. They might also recruit help from non-bioinformatics colleagues of theirs and see where they fail.

The software must be in working order for publication to go ahead and this should not involve the user having to fix or work around issues not properly documented by the authors.

- What are the plans for maintenance and development of the software? This should also be stated in the manuscript.

Major points -- manuscript:

- No proper comparison to existing solutions is performed, which makes it difficult to put the tool into perspective and to assess novelty. The authors only state that "If tools for scATAC-seq are publicly available, [...] these tools are dedicated to scATAC-seq and are not stand-alone applications" (which is a tautology and should be re-written). The authors demonstrate the use of their tool for scATAC-seq data – can the other tools not also be applied to scChIP-seq data? If not, the reasons should be clearly stated. At least one other Shiny tool exists that boasts support for scChIP-seq data: <https://zhiji.shinyapps.io/scrat/>. There are also standalone applications that support scATAC-seq data, e.g. scOrange (<https://singlecell.biolab.si/>). A more systematic and comprehensive comparison in terms of functionality, mode of implementation, usability, and limitations to all existing frameworks (including but not limited to the software quoted in the current manuscript) should be made. This should probably go with a Supplementary Figure or Table.

- "The pipeline is designed for high-throughput single-cell datasets with samples containing as low as 100 cells and with a minimum of 1,000 reads per cell" (l. 58ff.) This establishes a lower bound on the supported data, but what's the upper bound? As single-cell datasets grow larger and larger, can the software deal with that? What are the parameters that influence scalability?

Minor:

- The paper references the Docker Hub repository (pacomito/chromscape), which eases installation, but the link is missing on the GitHub documentation.

- Spell tools / resources correctly: SAMtools, bedtools, GENCODE, Benjamini-Hochberg (not "Hocheberg").

- "We haven't observed any batch effect between our experiments in our first study most probably because we were working with a single batch of hydrogel beads. No batch correction option is implemented in this version of the application." (l. 116ff.) I imagine the lack of batch correction / dataset integration methods (such as implemented in Seurat, LIGER, CONOS, etc.) will be major limitation for practical usefulness in future applications. The extent to which these problems pertain to scChIP-seq data is currently not clear and an evaluation seems beyond the scope of this manuscript (which is why I'm listing this as a minor point here, even though I consider it very important). Based on my experience with all other types of genomics data, however, I can only assume that batch effects will be pervasive. Do the authors have any plans of adding functionality to deal with this, e.g. by integrating methods such as the ones from Seurat/Signac, etc?

- The language needs some correcting, but I will leave details to the editors.

Point by point response to reviewers comments

Reviewer #1 (Remarks to the Author):

Prompsy, Kirchmeier et al. present ChromScape a web application for the analysis of single cell ChIP-seq or ATAC-seq datasets. The application is aimed to researcher without bioinformatic expertise. ChromScape implements common steps for the analysis and visualization of count matrices. The webapp is well made and in principle could be very helpful, however I think it is important to address these points before publication.

Major points:

1) *If the goal is to provide an end to end pipeline for people without scripting skills it is necessary to start from raw sequencing data and not from count data. These steps should be included in the current pipeline and integrated in the Shiny user interface. This can be easily accomplished by integrating the other library you have created: https://github.com/vallotlab/scChIPseq_DataEngineering.*

We agree with the reviewer that initial processing steps for single-cell experiments require scripting skills, but above all these require the usage of a computing cluster to manipulate the raw datasets. Raw reads (in FASTQ format) are extremely large files that can't be supported in terms of memory and treatment time on a personal computer. In addition, among single-cell epigenomic technologies, as for scRNA-seq methods, there are a variety of strategies for barcode and library design, that each require very different data engineering steps to demultiplex cells and align reads correctly. Concerning cell barcodes for example, they may be of different length and either generated by primer extension (snATAC-seq) or by split and pool strategies, using a combination of a known set of indexes (scChIP-seq). Altogether, dealing with raw counts appears to be too dataset-specific and requiring too much computing capabilities, to fit within a rather nimble Shiny/R application. We have chosen to modify ChromScape to take as input aligned reads in addition to formatted count tables in order to facilitate and enlarge its usage. Starting from aligned reads, where all design specificities have been accounted for, seems a good solution to keep the app as flexible and user-friendly as possible. In addition, aligned reads (in BAM and BED formats) are standard data formats for published data sets in GEO and other repositories.

To this end, we have implemented 2 additional functionalities that allow user to upload single-cell BAM files or single-cell BED files. The signal is now then aggregated into either genomic bins, peaks (the user must provide an additional BED file containing peak coordinates), or around gene transcription start sites (TSS) - the user can change the up/downstream region to take into account around TSS. We believe allowing a large variety of formats and letting the user test multiple way of counting (e.g. different bin sizes, bins vs peaks...) will greatly enlarge the usability of ChromScape. On a 32Gb Linux with 8 cores, reading & counting for 2,500 cells files into more than 60,000 50kbp-bins took: (i) from scBAM files : 15 minutes, not exceeding 16Gb of RAM used (depending on number of cores available), (ii) from scBED files: 9 minutes, not exceeding 11Gb of RAM used (depending on number of cores available).

In the manuscript, we have now made it clear (line 60) that ChromScape is an end-to-end pipeline for data analysis and not data engineering of single-cell epigenomic datasets.

2) *Although the goal is to provide a user-friendly graphical application it is not clear what is the quality of the results produced. To this end it is important to compare the results obtained with this pipeline with recent pipelines even though they don't provide a web-based interface.*

- *For ChIP-seq data, please compare your results with methods also based on count matrices like EpiScanpy (<https://europepmc.org/article/ppr/ppr80155>) or Seurat (https://satijalab.org/seurat/v3.0/atacseq_integration_vignette.html).*

- *For ATAC-seq data in addition to EpiScanpy several methods have been proposed. A systematic comparison is presented in Chen et al 2019 (<https://genomebiology.biomedcentral.com/articles/10.1186/s13059-019-1854-5>). From their analysis it is clear that it is not necessary to compare ChromScape with all the available recent methods but only with the top performing i.e. SnapATAC, Cusanovich 2018 and CisTopic. Without this comparison it is hard to assess the quality of the proposed results.*

In line with comments from all reviewers, we have now compared the performance of ChromScape to existing pipelines. We have tested four pipelines without web-based interface - EpiScanpy along with the three best performing methods for scATAC-seq methods (according to the benchmark by Chen et al. 2019). To date, performing a benchmark for scChIP-seq data analysis remains a challenge given the very low availability of reference datasets, especially for repressive histone marks such as H3K27me3. Here, we mixed together several of our H3K27me3 scChIP-seq datasets to be able to evaluate the performance of each pipeline according to the known ground truth. We mixed breast tumor cells from our HBCx22 PDX model and Jurkat and Ramos cells from our previous study (Grosselin et al. 2019), and cells from a breast cancer cell line MDA-MB-468 (previously unpublished data, now submitted to GEO, GSE152502). Results are now included in Figure 2 of the manuscript. All cells have been processed with the same design of beads, and Jurkat & Ramos cells were processed using the exact same batch of beads within one microfluidics experiment (as explained in Grosselin et al. 2019). We

show that all computational methods manage to separate the 4 cell types, with ChromSCape being in the top-performing methods. We are aware that this comparison is somewhat limited by the reference dataset, for which cells might be easily segmented; yet to our knowledge it is the only dataset with ground truth for histone modifications. In the future, it will be interesting to perform again such benchmarking with more complex datasets, mixing for example several normal subtypes with minor differences in epigenomic landscapes.

Regarding the analysis of scATAC-seq datasets, we have analyzed with ChromSCape datasets from patients & cell lines assembled from Buenoroastro et al. 2015, Corces et al. 2016, Schep et al. 2017. This combined dataset was used in the SnapATAC paper (see Figure 3a in Fang et al. 2019) in a similar manner as we did in this manuscript.

3) *"No batch correction option is implemented in this version of the application." Explain the implications of this, warn the users and suggest some alternatives to remove batch effects using available packages. Alternatively, consider including a simple method for batch correction in ChromSCape, for example, the method developed by the Marioni lab (<https://www.ncbi.nlm.nih.gov/pubmed/29608177>) based on mutual nearest neighbors.*

We have now implemented in ChromSCape a batch correction feature, based on the FastMNN algorithm as suggested by the reviewer. If necessary, the user can now specify batch information for each set of cells. In such case, reduced feature space is corrected for batch effect for further downstream analysis. Regarding differential analysis, to account for batch effect, we have implemented the pairwiseWilcox function from the *scr*an package, setting the batch of origin as a 'blocking level' for each cell.

To test this functionality, we have run ChromSCape on H3K27me3 scChIP-seq datasets from human cells originating from the same PDX model (HBCx95) but using two different batches of beads, expecting a batch effect. The first batch corresponds to cells from our previous study (Grosselin et al., 2019), and we have now performed an additional scChIP-seq experiment on a biological replicate of HBCx95 using a novel batch of beads (now submitted to GEO, GSE152502). To control for over-correction, we have included cells from a resistant tumor (HBCx-95-CapaR, Grosselin et al. 2019) processed with one of the batches, which are expected to display a different epigenome than cells from HBCx-95. Without any batch correction, cells group according to version of the beads rather than cell identity, whereas following FastMNN-based correction, cells group according to sample of origin. We have now integrated this analysis in Supplementary Figure 2 and in the manuscript lines 305 to 317.

4) *Please provide some information on memory requirement and running time for the datasets presented. Also, what is the maximum count matrix size a user can analyze on a personal computer with 8 or 16GB or memory?*

On a MacBookAir7,2 Intel Core i5 1,6 GHz with 2 cores and 4 Gb of RAM, ChromSCape was able to read simultaneously 11 count matrices totalling 3.9 Gb, corresponding to more than 29,000 cells. The filtering step, keeping ~10,000 cells, and dimension reduction & clustering as well as display of heatmap took less than 60 minutes. The differential analysis & gene sets analysis took another additional 60 minutes. The advantage of ChromSCape is that the user only needs to upload & process count matrices once, the result of these steps is then stored and can be loaded automatically to start directly with differential analysis and gene set enrichment.

5) *The peak analysis is disconnected from the rest, since the tool takes in input count matrices, however for this step is necessary to obtain bam files and to have experience with command line tools and scripting. As mentioned before to make ChromSCape truly useful to non-expert, it is necessary to start from raw sequencing data.*

As mentioned in our answer to the first comment above, ChromSCape can now start from aligned files (single cell BAM or BED files); starting from raw fastq files did not seem feasible within a lean RShiny App, considering the diversity of technological design and the size of raw sequencing files.

Regarding the peak calling, we have now enhanced our peak calling feature to make it as straightforward as possible, the user now only has to select the folder containing the BAM files, and the function will automatically separate cells by cluster and run the peak calling. Unfortunately, there is currently no peak calling tool implemented in R - as MACS2 - that runs without the need of a matched input (which we do not have for single cell epigenomic datasets). Therefore we did not find a solution to implement the peak calling step within the app directly. Yet, we see the peak calling step as an option, it not mandatory, and even more, useless when the aggregation of signal has been done initially on defined genomic regions, as ChromSCape now enables it.

6) *The support for ATAC-seq data is very primitive and limited. The assumption again is to start from a count matrix and some custom scripts were necessary to proceed with the analysis "The count matrix of reads in 188 peaks was downloaded from GEO accession number GSE99172, split into distinct matrices for 189 each sample and formatted to be accepted as input by ChromSCape." If you want to support scATAC-seq datasets it is*

necessary to include the preprocessing steps to get a matrix from raw reads.

We have now modified ChromSCape so that it can automatically handle a 'combined matrix' regrouping cells of various conditions or cell types like in GSE99172. ChromSCape automatically detects which cells belong to which sample based on cell names. ChromSCape now also accepts .csv files. The user only has to download the count matrix at GSE99172, unzip it and upload it in ChromSCape, checking the 'contains multiple sample' box and precisising the number of samples (here 20 samples x 96 cells = 1,920 cells). The application will automatically recognize conditions based on cell names.

Minor points:

1) Line 286: extra space in "Ad ditionally"

We have now removed this extra space.

2) Windows support? In the docker version clear instructions are provided for Linux/Mac OS, however no instructions are provided for Windows. This is a missing opportunity given that Docker is also supported on Windows and a simple modification can be provided to support also this platform.

ChromSCape is supported on windows, except for the optional peak calling part which requires MACS2 & SAMtools, two tools not available on windows. We have now successfully installed from github and used ChromSCape package on several Windows machines.

Reviewer #2 (Remarks to the Author):

Recent technological advancement enables epigenetics analysis at a single-cell level, but analysis tools dedicated for sparse single-cell epigenome datasets are currently lacking and thus desirable. In this study, Prompsy et al. developed ChromSCape, a GUI application that is designed for single-cell data and allows e.g. identification of subpopulations of cells based on epigenome states. Using ChromSCape, the authors showed the ability of their software that accepts count matrix of genomic bins (cells x bins) and automates the following data analysis steps including filtering outlier cells, lower-dimensional representation, clustering, calling peaks and differential enriched regions. They performed demonstrations on scChIP-seq data from their previous report as well as scATAC-seq data used for another scATAC-seq analytical tool chromVar, showing ChromSCape worked well on both datasets. Thus, this work has a potential to liven the blooming field of single-cell epigenomic by providing a user-friendly analysis tool. At this stage, however, the software seems immature and specifically dedicated to their own droplet-based scChIP-seq technology. This manuscript merely introduces their data analysis tips and a combo of BioConductor tools, rather than proposes novel computational/statistical ideas and/or technical advancements in the sparse and high-dimensional single-cell epigenomic data analysis. Overall, although ChromSCape could be a valuable tool for single-cell epigenome analysis, the manuscript in its current form does not necessarily need to be published on the journal Nature Communications and thus may be suitable for other more specific journals.

Specific points

1: *The authors stated that "With ChromSCape (Fig. 1), we propose a user-friendly, step-by-step and customizable R/Shiny application to analyze sparse single-cell histone modifications datasets. (line 54)". Setting up ChromSCape, however, appears to still require high computational skills, especially for general biologists. More importantly, installing the application does not work in my brief trial (both R environment and docker container, a point related to path resolving problem after uploading count matrix). This implies this software has not been well tested.*

We have now changed our application into an R package, which can be installed with a one-line command line in the R console, `install_github("vallolab/ChromSCape")`. This new version of the application, with multiple new functionalities - including new input formats, batch correction, UMAP visualization, multiple differential testing - has been tested by several colleagues, both biologists & bioinformaticians, to make sure no bugs is encountered during installation & use of ChromSCape (tested on 2 windows machine, 4 mac, and 2 linux). All dependencies are installed automatically when running the installation command and the app is be ready to launch as soon as the installation is finished. Additionally, we have published a demo server of the application at <https://vallotlab.shinyapps.io/ChromSCape/> for users to take a virtual tour of the app before installing it if necessary.

2: *ChromSCape does not provide a general guideline for a wide variety of single-cell epigenomic technologies recently developed, such as CUT&Tag, scChIL, scChIC and CoBATCH. It seems the authors merely follow droplet-based scRNA-seq data analysis (e.g., "In order to efficiently remove outlier cells from the analysis, e.g. cells with excessively high or low coverage, the user sets a threshold on a minimum read count per cell and the upper percentile of cells to remove. (line 92)"). The authors should provide a universal exploratory data analysis platform for e.g. deciding filtering threshold, optimizing bin-size, correction of unwanted variations and batch-effects. Otherwise its applicability and/or robustness on many other single-cell epigenomic datasets have not sufficiently been demonstrated.*

Following the reviewer's comment, we have now implemented a series of functionalities for ChromSCape to become a universal exploratory tool for single cell epigenomic datasets, both chromatin accessibility and histone modification datasets, whatever the technology. To do so, ChromSCape now allows as input standard single-cell formats for aligned reads rather than only specific formatted count tables. Users can now upload single-cell BAM files or single-cell BED files. The user can choose to aggregate the signal into either genomic bins, peaks (the user must provide an additional BED file containing peaks), or into regions around gene TSS (the user can change the up/downstream region to take around the TSS). Allowing a large variety of inputs and letting the user test multiple way of counting (e.g. different bin sizes, bins vs peaks...) will indeed greatly enlarge the usability of ChromSCape.

We have also evaluated the usability of ChromSCape for other types of single-cell histone modification data obtained by other technologies than scChIP-seq. We have analyzed two public datasets of scCUT&Tag and scChIC-seq targeting H3K27me3 and H3K4me3 marks respectively. ChromSCape now facilitates the analysis of such public dataset as the user only has to download single-cell BED files and directly input them in the application, specify the number of samples and can choose the partition into which one aggregates the signal. We have now included these analysis in Supplementary Figure 1a, and in the manuscript lines 318 to 328. For both datasets, ChromSCape efficiently separates cells as in the original studies. We measured an ARI of 0.976 for K562 & H1 cells for scCUT&Tag, and for the scChIC-seq dataset where no ground truth was available, we

recapitulate the segmentation in 7 clusters on the UMAP representation, as found by the authors in their initial study.

We have now implemented in ChromSCape a batch correction feature, based on the FastMNN algorithm as suggested by the reviewer. If necessary, the user can now specify batch information for each set of cells. In such case, reduced feature space is corrected for batch effect for further downstream analysis. Regarding differential analysis, to account for batch effect, we have implemented the pairwiseWilcox function from the scan package, setting the batch of origin as a 'blocking level' for each cell.

To test this functionality, we have run ChromSCape on H3K27me3 scChIP-seq datasets from human cells originating from the same PDX model (HBCx95) but using two different batches of beads, expecting a batch effect. The first batch corresponds to cells from our previous study (Grosselin et al., 2019), and we have now performed an additional scChIP-seq experiment on a biological replicate of HBCx95 using a novel batch of beads (now submitted to GEO, GSE152502). To control for over-correction, we have included cells from a resistant tumor (HBCx-95-CapaR, Grosselin et al. 2019) processed with one of the batches, which are expected to display a different epigenome than cells from HBCx-95. Without any batch correction, cells group according to version of the beads rather than cell identity, whereas following FastMNN-based correction, cells group according to sample of origin. We have now integrated this analysis in Supplementary Figure 2 and in the manuscript lines 305 to 317.

3: (Calling differentially enriched regions) Comparing one specified cluster with all the others seems less sensitive for detecting candidates because of the large variability of out-of-cluster cells. Providing options to test all possible pairs of clusters will be useful for users.

Following the reviewer's suggestion, we have now implemented in ChromSCape a pairwise testing option in addition to one-vs-rest differential testing to enable the user to perform every possible comparison between clusters.

4: (Statistical modeling for categorical variables) Why do the authors prefer non-parametric (Wilcoxon rank sum) tests over Poisson or Negative binomial modeling of count data? If recommended, the authors should compare these different statistical approaches with regard to sensitivity/specificity.

We had chosen the non-parametric Wilcoxon rank sum test to avoid making any assumptions on the distribution of scChIP-seq data. In our hands, the distribution of scChIP-seq data is even more zero-inflated than scRNA-seq, due to the inherent experimental limitations, and has a lower dynamic range. In addition, a recent thorough benchmark of differential analysis methods for single-cell data have shown that Wilcoxon test is among the best performing tests together with parametric tests (Soneson & Robinson 2018).

Yet, now that ChromSCape takes in input several types of single-cell epigenomic datasets, we have decided to leave the choice to the users depending on the distribution of each studied dataset. Following the reviewer's suggestion, we have now implemented in ChromSCape an option to run parametric differential testing using 'edgeR' package (McCarthy et al., 2012) that assumes that the signal follows a negative binomial distribution. Such approach has been shown to be an adapted modeling strategies for single-cell RNA datasets (Soneson & Robinson 2018). We have just started to work on the development of a specific statistical framework to model scChIP-seq datasets in collaboration with Jean-Philippe Vert at Google Lab, and will implement it in ChromSCape in the future.

Reviewer #3 (Remarks to the Author):

Prompsy and colleagues describe in their manuscript a software (called ChromSCape) for the analysis of single-cell ChIP-seq data and illustrate the use with data from their recent publication (Grosselin et al. 2019, Nat. Genet.). Overall, I believe the chosen examples serve the illustrative purpose and I appreciate the attempt to create user-friendly software for single-cell data analysis, but I believe the solution falls completely short of providing a truly user-friendly tool in its current form. The authors also need to add a proper comparison to the state-of-the-art. If these issues can be resolved / improved upon, I would be supportive for publication in Nature Communications.

Major points -- software:

- *The tool is primarily targeted at experimental biologists with limited bioinformatics skills (“[to] enable users without any bioinformatics skills [...] run the application in a web browser”). I very much doubt that the average target user will be able to install the software in its basic setup (R, bedtools, command line execution) and even less so with additional packages required for advanced analysis (SAMtools, bedtools, MACS2, for peak calling I. 144ff.). For me, the installation script asking me to downgrade my Bioconductor release, which I of course chose not to do to avoid conflicts with other projects. Setting up an R install just for ChromSCape is a big ask. Really, the authors should implement ChromSCape as a Bioconductor package which will force them to mind dependencies and versioning.*

- *Providing a Docker image with pre-installed software requirements is a good thought in order to ease the installation burden (this is what I chose to do after the problems I described above), however, this first requires Docker to be installed and admin rights on the computer, plus being comfortable with setting up directories and changing permissions –again considerable stumbling stones for many prospective users.*

All installation problems can probably be managed with additional time investment and help of computer-savvy colleagues, but a quick and easy way to test and trial the software is required for potential users to evaluate whether it’s worth to go through all that trouble. I would therefore highly recommend setting up a public demo server with the application that allows immediate execution with a trial dataset.

- *Even after installing the Docker version of the software from Docker Hub and following the instructions on the GitHub page about creating directories and running as root, I faced difficulties using it (I used Ubuntu Linux). Uploading new datasets (I used the count matrices provided on figshare) consistently resulted in disconnection from the server. I then tried to download the precompiled analysis (also provided by the authors on figshare) and tried putting it in the Data_ChromSCape folder (no detailed instructions provided what else to do with it) – I could not see it from the running application. I gave up at this point, having spent a considerable amount of time without being able to try the software.*

It seems to be that these kind of issues should be resolved by the authors by some proper software testing. They should install the software following their own instruction on a “clean” computer – i.e. one that is not already set up with all their commonly used tools and configuration. They might also recruit help from non-bioinformatics colleagues of theirs and see where they fail. The software must be in working order for publication to go ahead and this should not involve the user having to fix or work around issues not properly documented by the authors.

We agree the first version of ChromSCape needed too much pre-requisite for a non-computational biologist to manage installation. We have now removed the Docker version of the software which needed too much manual commands that could vary from an operating system to another, as well as changing the ownership of folders. We have now implemented ChromSCape as a stand-alone R package, requiring R version 3.6. We have replaced every call of the BEDTools software by the R functions implemented in the GenomicRanges Bioconductor package, and the only non-R dependencies are now in the optional peak calling step, and the user is warn if the non-R dependencies are not found. The installation through the single command line “devtools::install_github” will take care of installing all the dependencies required to run ChromSCape thus facilitating the installation. Following the reviewer’s recommendation, we had our application tested by several computer biologists and biologists to make sure the installation and use of the application was smooth under different systems (Linux, Mac OS, Windows) with R3.6.

In addition, following the reviewer’s recommendation, we have also now implemented a demo server available at <https://vallotlab.shinyapps.io/ChromSCape/>. You can find on the demo server, a trial dataset for users to discover ChromSCape before installing it and running locally. Also, we have added a ‘guided-tour of the application’ when the user launches the app for the first time, as well as help buttons for all the major modules for the user to understand what the module exactly does. Additionally we have written a pdf guide that explains how to use the application, available at https://vallotlab.github.io/ChromSCape/ChromSCape_guide.html. We believe that these changes will make ChromSCape more user-friendly and accessible to all researchers.

- *What are the plans for maintenance and development of the software? This should also be stated in the manuscript.*

Formatting ChromSCape into an R package will help us maintain it. We also plan on further developing the application by keeping ChromSCape up to date with the fast evolving functionalities and technologies developed in the single-cell epigenomic field and developing specific algorithms to deal with single cell epigenomic datasets. For instance, we have just started to work on the development of a specific statistical framework to

model scChIP-seq datasets in collaboration with Jean-Philippe Vert at Google Lab, and will implement it in ChromSCape in the future to enhance its ability to detect differentially enriched regions between cells.

Major points -- manuscript:

- No proper comparison to existing solutions is performed, which makes it difficult to put the tool into perspective and to assess novelty. The authors only state that "If tools for scATAC-seq are publicly available, [...] these tools are dedicated to scATAC-seq and are not stand-alone applications" (which is a tautology and should be re-written). The authors demonstrate the use of their tool for scATAC-seq data – can the other tools not also be applied to scChIP-seq data? If not, the reasons should be clearly stated. At least one other Shiny tool exists that boasts support for scChIP-seq data: <https://zhiji.shinyapps.io/scrat/>. There are also standalone applications that support scATAC-seq data, e.g. scOrange (<https://singlecell.biolab.si/>). A more systematic and comprehensive comparison in terms of functionality, mode of implementation, usability, and limitations to all existing frameworks (including but not limited to the software quoted in the current manuscript) should be made. This should probably go with a Supplementary Figure or Table.

In line with comments from all reviewers, we have now compared the performance of ChromSCape to existing pipelines with or without interface. We have first tested four reference pipelines (without graphic interface) - EpiScanpy along with the three best performing methods for scATAC-seq methods (according to the benchmark by Chen et al. 2019). To date, performing a benchmark for scChIP-seq data analysis remains a challenge given the very low availability of reference datasets, especially for repressive histone marks such as H3K27me3. Here, we mixed together several of our H3K27me3 scChIP-seq datasets to be able to evaluate the performance of each pipeline according to the known ground truth. We mixed breast tumor cells from our HBCx22 PDX model and Jurkat and Ramos cells from our previous study (Grosselin et al. 2019), and cells from a breast cancer cell line MDA-MB-468 (unpublished dataset, now submitted to GEO, GSE152502). Results are now included in Figure 2 of the manuscript. All cells have been processed with the same design of beads, and Jurkat & Ramos cells were processed using the exact same batch of beads within one microfluidics experiment (as explained in Grosselin et al. 2019). We show that all computational methods, including ChromSCape, manage well to separate the 4 cell types. We are aware that this comparison is somewhat limited by the reference dataset, for which cells might be easily segmented, yet to our knowledge it is the only dataset with ground truth for histone modifications. In the future, it will be interesting to perform again such benchmarking with more complex datasets, mixing for example several normal subtypes with minor differences in epigenomic landscapes.

Following the reviewer's comment, we have also compared ChromSCape to two applications with graphic interface destined to biologists & bioinformaticians: scOrange (<https://singlecell.biolab.si/>) & SCRAT (<https://zhiji.shinyapps.io/scrat/>), results are also included in Figure 2 of the manuscript.

ScOrange is a stand-alone platform allowing researchers to create their own workflow to analyze single-cell datasets by proposing a great variety of 'modules', such as loading, filtering, reducing the dimensions by PCA or t-SNE, and producing quality scatter plots and heatmaps. However, the users need to have knowledge of what analytical steps are required to identify their cell populations, which is non trivial in the case of single-cell epigenomic datasets. When running with scOrange the analysis of an in-silico mix of H3K27me3 scChIP Jurkat, Ramos, HBCx22 & MDA-MB-468 with one of the default workflows ('Clustering & Cluster Analysis'), cells types were not separated in reduced space, but only after optimizing these steps (see Fig. 2c, 'Optimized' vs 'Default'). The tool is currently more dedicated to scRNA-seq, as no 'template' workflow exists for single-cell epigenomic data (scATAC or scChIP). In addition, there is no way to link genomic regions to gene annotation, in order to properly interpret the cell clusters. While we think that scOrange is a great tool for researcher that are ready to spend more time on the software to create dedicated new workflows, the application is currently not 'hands-on' to analyze and make sense of epigenomic single cell datasets.

SCRAT is a R/Shiny package to analyze single-cell epigenomic data, similar to ChromSCape. SCRAT takes BAM files as input and offers the user to count on various features : Transcription Factor Motifs, co-regulated DNase Hypersensitivity sites on ENCODE, Genes or custom features (such as peaks called beforehand by the user). The user interface and the visualization is however quite limited as the user is not informed of the cell labels or conditions for example. When running SCRAT with default parameters on the *in-silico* mix, we didn't manage to separate cells types at all. When optimizing parameters, we obtained 4 clusters, but with no information on the sample of origin of the cells. In addition, as for scOrange, the interpretation of the data is limited, with no pathway enrichment analysis to interpret clusters.

Thanks to the different comments from reviewers, we believe ChromSCape now has a series of functionalities to become a universal exploratory tool for single cell epigenomic datasets. It now allows new input formats, batch correction, UMAP visualization, multiple differential testing possibilities, that will greatly enlarge its usability. In addition, it can handle both chromatin accessibility and histone modification datasets whatever the technology (see Sup. Fig. 1). We have indeed evaluated the usability of ChromSCape for other types of single-cell histone modification data obtained by other technologies than scChIP-seq. Analyzing two public datasets of scCUT&Tag and scChIC-seq targeting H3K27me3 and H3K4me3 marks respectively, we have observed that ChromSCape can efficiently separates cells as in original studies.

- *"The pipeline is designed for high-throughput single-cell datasets with samples containing as low as 100 cells and with a minimum of 1,000 reads per cell" (l. 58ff.) This establishes a lower bound on the supported data, but what's the upper bound? As single-cell datasets grow larger and larger, can the software deal with that? What are the parameters that influence scalability?*

On a MacBookAir7,2 Intel Core i5 1,6 GHz with 2 cores and 4 Gb RAM, ChromSCape was able to read in 11 matrices totalling 3.9 Gb, more than 29,000 cells. The filtering to keep ~10,000 cells, dimension reduction & clustering as well as display of heatmap took less than 60 minutes, the differential analysis & gene sets analysis took another additional 60 minutes. The advantage of ChromSCape is that the user only needs to upload & process the matrices once and can then load automatically the saved analyses.

Minor:

- *The paper references the Docker Hub repository (pacomito/chromscape), which eases installation, but the link is missing on the GitHub documentation.*

This will be removed since the code is now embedded in a R package and all dependencies will automatically be installed when running the installation command.

- *Spell tools / resources correctly: SAMtools, bedtools, GENCODE, Benjamini-Hochberg (not "Hocheberg").*

We have now corrected the spelling in the text.

- *"We haven't observed any batch effect between our experiments in our first study most probably because we were working with a single batch of hydrogel beads. No batch correction option is implemented in this version of the application." (l. 116ff.) I imagine the lack of batch correction / dataset integration methods (such as implemented in Seurat, LIGER, CONOS, etc.) will be major limitation for practical usefulness in future applications. The extent to which these problems pertain to scChIP-seq data is currently not clear and an evaluation seems beyond the scope of this manuscript (which is why I'm listing this as a minor point here, even though I consider it very important). Based on my experience with all other types of genomics data, however, I can only assume that batch effects will be pervasive. Do the authors have any plans of adding functionality to deal with this, e.g. by integrating methods such as the ones from Seurat/Signac, etc?*

We have now implemented in ChromSCape a batch correction feature, based on the FastMNN algorithm as suggested by the reviewer. If necessary, the user can now specify batch information for each set of cells. In such case, reduced feature space is corrected for batch effect for further downstream analysis. Regarding differential analysis, to account for batch effect, we have implemented the pairwiseWilcox function from the scran package, setting the batch of origin as a 'blocking level' for each cell.

To test this functionality, we have run ChromSCape on H3K27me3 scChIP-seq datasets from human cells originating from the same PDX model (HBCx95) but using two different batches of beads, expecting a batch effect. The first batch corresponds to cells from our previous study (Grosselin et al., 2019), and we have now performed an additional scChIP-seq experiment on a biological replicate of HBCx95 using a novel batch of beads (now submitted to GEO, GSE152502). To control for over-correction, we have included cells from a resistant tumor (HBCx-95-CapaR, Grosselin et al. 2019) processed with one of the batches, which are expected to display a different epigenome than cells from HBCx-95. Without any batch correction, cells group according to version of the beads rather than cell identity, whereas following FastMNN-based correction, cells group according to sample of origin. We have now integrated this analysis in Supplementary Figure 2 and in the manuscript lines 305 to 317.

- *The language needs some correcting, but I will leave details to the editors.*

REVIEWERS' COMMENTS:

Reviewer #1 (Remarks to the Author):

I acknowledge all the hard work that the authors have put in revising this manuscript and I am overall satisfied with the point by point response, however I feel that the ChromScape software has not been tested sufficiently and therefore not ready for publication.

I tried on a fresh installation of R 3.6.3 (Windows 10 machine) to follow their instructions, however the installation failed with the following error:

```
Error: Failed to install 'ChromSCape' from GitHub:  
(converted from warning) dependency 'XML' is not available
```

The command I have used is, as instructed in the documentation:

```
if (!requireNamespace("devtools", quietly = TRUE)){  
install.packages("devtools")  
}  
devtools::install_github("vallotlab/ChromSCape")
```

I tried to manually install the XML library but without success:

```
Warning in install.packages :  
package 'XML' is not available (for R version 3.6.3)
```

Also I suggest to provide again the Docker image (it was excluded after the revision). If you install macs2 and samtools within the docker image, Windows users can still have the option to perform the peak calling procedures and the software will still work and it will be easier to maintain long term.

Reviewer #2 (Remarks to the Author):

I appreciate the author's many efforts on this fairly improved version of this manuscript (benchmarks and supports for other sc technologies) and several new features to ChromScape. However, I could not find either of the conceptual advance of sc epigenome analysis methods (novelty) nor completeness of the software (technically sound) in this stage.

I have tried to launch the software but had faced the following issues. I hope these comments are useful for you and It should be solved prior to open the software.

Technical Comments

The installation was succeeded using dev tools, however, some errors that seem to be related to dependency exists.

```
```\n
```

```
Error in : 'calculateQCMetrics' is defunct.
Use 'perCellQCMetrics' instead.
```

```
```\n
```

After managing this issue by hand, the next error shows up.

```
```\n
```

```
Error in filter_scExp: is(scExp, "SingleCellExperiment") is not TRUE
```

...  
I could not proceed next to the Filtering step.

Since ChromSCape is largely dependent on other BioC packages, I recommend submitting your packages to BioConductor to resolve such dependency issues before publication.

Reviewer #3 (Remarks to the Author):

Prompsy et al. submit a revised version of their manuscript about ChromSCape, a web tool for single-cell epigenomics data analysis.

I had two main concerns about the initial submission: usability problems with the software and a missing comparison to other solutions in the manuscript. I believe the authors have made a good attempt at improving on both points: the software is now available as an R package and a demo server has been provided (which I have briefly tried out and both seem to work), and the manuscript contains a satisfactory comparison to other software. I also note that another of my minor points, which was also picked up by other reviewers, namely batch correction, has also been addressed. I am therefore quite happy with the revised paper and would be in favor of publication in Nature Communications.

There are a few mistakes in the text, which I trust the editors will help to resolve. I also have a few small comments mostly concerning the figures / plots produced by the software and practical improvements:

1. Plot quality in Fig. 2c is rather poor with tiny, illegible text. This might be ok/necessary if the plots are direct outputs of the tools that were used for comparison – which I think is the case – but please state this explicitly in the legend.
2. Fig. 3b/d: Colors used for “Percentage of cells assigned to cluster” are quite difficult (impossible) to interpret exactly. I would suggest to use a more informative color scale, possibly broken up in steps of 10 or 5 percentage points. Perhaps also numbers could be overlaid on the heatmaps.
3. Fig. 3a/c: Different shades of dark/light color are used for “replicates” of the same sample group. However, at the same time, transparency is used to show overlapping points. This can make it difficult to distinguish points belonging to each sample. I also note that light/dark color are used inconsistently for blasts (SU353 light, SU070 dark) and LSCs (SU353 dark, SU070 light). My comments about colors vs. transparency also pertain (even more strikingly) to Fig. 4b/c.
4. Fig. 4a: “Count” is ambiguous. The label should be replaced by “Read count”, “Number of mapped reads”, or something similar.
5. Fig. 4d: No color key is provided.
6. Fig. 4f: Not clear what “Count” means in these plots.
7. Fig. 4h: Not entirely clear how “top 15” pathways were ranked – by adjusted p-value? The plot would probably be easier to read if rotated clockwise.
8. “Single-cell H3K27me3 enrichment levels can be visualized for genes within the top 100 most significant differential regions using a t-SNE.” (l. 188ff) – I don’t understand how a t-SNE (low-dimensional projection) is used for visualization of enrichments. Does this refer to overlaying H3K27me3 counts at selected genes onto a t-SNE plot, as in Fig. 4f?
9. I note a competing interests statement has been added without disclosure of the names of all parties involved. I’m not sure what the journal’s policy is on this.
10. Support for multiple data inputs has been provided. Since data files can become quite large, a straightforward improvement (I would argue a necessary improvement) would be to add support for gzipped files (with a gzip connection wrapper). They can be largely handled in the same way as uncompressed text files but save on disk space and data transfer.
11. Why is there a maximum upload size even when I’m trying to upload to my local server? I couldn’t upload data obtained from Satpathy et al 2019 (GSE129785).
12. I did not manage to select files using the multi-file selector for “Index, Peak & Barcode files”,

"Single-cell BAM files", or "Single-cell BED files" inputs when using Google Chrome browser: I get a popup dialog and can browser files, but I did not find a way to select files. Perhaps I missed something (if so, can it be made more obvious?). Why not just use the standard file selector as for "Count matrix(ces)"?

Finally an idea for future improvements: The authors perform pathway enrichment analysis using MSigDB. There are a number of tools for performing functional enrichment analysis either based on associating genomic regions with genes (e.g., GREAT, McLean et al (2010), <http://great.stanford.edu/>) or based on overlaps with "known" genomic regions (e.g. coloc-stats, Simovski et al (2018), <https://hyperbrowser.uio.no/coloc-stats/>; LolaWeb, Nagraj et al (2018), <http://lolaweb.databio.org/>). The authors may consider inclusion / linking out to these tools (many also come as web services) as a future improvement for ChromSCape.

## Reviewer's comments:

### Reviewer #1 (Remarks to the Author):

*I acknowledge all the hard work that the authors have put in revising this manuscript and I am overall satisfied with the point by point response, however I feel that the ChromScape software has not been tested sufficiently and therefore not ready for publication.*

*\* I tried on a fresh installation of R 3.6.3 (Windows 10 machine) to follow their instructions, however the installation failed with the following error:*

*Error: Failed to install 'ChromSCape' from GitHub:  
(converted from warning) dependency 'XML' is not available*

*The command I have used is, as instructed in the documentation:*

```
if (!requireNamespace("devtools", quietly = TRUE)){
install.packages("devtools")
}
devtools::install_github("vallotlab/ChromSCape")
```

*I tried to manually install the XML library but without success:*

*Warning in install.packages :  
package 'XML' is not available (for R version 3.6.3)*

We had tested ChromSCape installation on over 10 machines (both Mac and Windows), yet did not encounter such error. As suggested by two reviewers, to prevent any installation problems due to package dependencies, and ensure a robust and stable way to install our application, we have now submitted our package for the next BioConductor release (<https://github.com/Bioconductor/Contributions/issues/1642>). To do so, we have adapted our package to R version 4.0 and current devel of Bioconductor (version 3.12). Our package has passed the Bioconductor checks and is currently under review for the next release, on **Wednesday October 28th**.

*\* Also I suggest to provide again the Docker image (it was excluded after the revision). If you install macs2 and samtools within the docker image, Windows users can still have the option to perform the peak calling procedures and the software will still work and it will be easier to maintain long term.*

We created a Docker Image for ChromSCape containing Bioconductor devel version 3.12 and R.4.0.2 plus samtools and MACS2 for long-term maintenance and optimal user experience on Windows (<https://hub.docker.com/r/pacomito/chromscape>). The command line needed to launch the docker container and the application is available on the github page.

## Reviewer #2 (Remarks to the Author):

*I appreciate the author's many efforts on this fairly improved version of this manuscript (benchmarks and supports for other sc technologies) and several new features to ChromSCape. However, I could not find either of the conceptual advance of sc epigenome analysis methods (novelty) nor completeness of the software (technically sound) in this stage.*

*\* I have tried to launch the software but had faced the following issues. I hope these comments are useful for you and It should be solved prior to open the software.*

*Technical Comments*

*The installation was succeeded using dev tools, however, some errors that seem to be related to dependency exists.*

...

*Error in : 'calculateQCMetrics' is defunct.*

*Use 'perCellQCMetrics' instead.*

...

*After managing this issue by hand, the next error shows up.*

...

*Error in filter\_scExp: is(scExp, "SingleCellExperiment") is not TRUE*

...

*I could not proceed next to the Filtering step.*

*Both errors were due to a change in the Bioconductor package 'batchelor' in newest versions of Bioconductor.*

*Since ChromSCape is largely dependent on other BioC packages, I recommend submitting your packages to BioConductor to resolve such dependency issues before publication.*

*As suggested by two reviewers, to prevent any installation problems due to package dependencies, and ensure a robust and stable way to install our application, we have now submitted our package for the next BioConductor release (<https://github.com/Bioconductor/Contributions/issues/1642>). To do so, we have adapted our package to R version 4.0 and current devel of Bioconductor (version 3.12). Our package has passed the Bioconductor checks and is currently under review for the next release, on **Wednesday October 28th**.*

## Reviewer #3 (Remarks to the Author):

Prompsy et al. submit a revised version of their manuscript about ChromSCape, a web tool for single-cell epigenomics data analysis.

*I had two main concerns about the initial submission: usability problems with the software and a missing comparison to other solutions in the manuscript. I believe the authors have made a good attempt at improving on both points: the software is now available as an R package and a demo server has been provided (which I have briefly tried out and both seem to work), and the manuscript contains a satisfactory comparison to other software. I also note that another of my minor points, which was also picked up by other reviewers, namely batch correction, has also been addressed. I am therefore quite happy with the revised paper and would be in favor of publication in Nature Communications.*

*There are a few mistakes in the text, which I trust the editors will help to resolve. I also have a few small comments mostly concerning the figures / plots produced by the software and practical improvements:*

*1. Plot quality in Fig. 2c is rather poor with tiny, illegible text. This might be ok/necessary if the plots are direct outputs of the tools that were used for comparison – which I think is the case – but please state this explicitly in the legend.*

We have now stated explicitly in the legend of Figure 2 that panel c is a direct snapshot of other tools used for comparisons: “Snapshots from scOrange and SCRAT applications”.

*2. Fig. 3b/d: Colors used for “Percentage of cells assigned to cluster” are quite difficult (impossible) to interpret exactly. I would suggest to use a more informative color scale, possibly broken up in steps of 10 or 5 percentage points. Perhaps also numbers could be overlaid on the heatmaps.*

We have modified the color scale and added breaks every 10 percentage points in Figure 3.

*3. Fig. 3a/c: Different shades of dark/light color are used for “replicates” of the same sample group. However, at the same time, transparency is used to show overlapping points. This can make it difficult to distinguish points belonging to each sample. I also note that light/dark color are used inconsistently for blasts (SU353 light, SU070 dark) and LSCs (SU353 dark, SU070 light). My comments about colors vs. transparency also pertain (even more strikingly) to Fig. 4b/c.*

In Figure 3a/c, we grouped all replicates under the same color, in order to avoid having too many colors in our panel (already 12), and assuming that the colors of replicates would be hard to differentiate from one another.

For the second part of the comment, the order of the legend labels in Fig 4b was indeed misleading (SU353, SU070, SU070, SU353). We changed that so that the order is now clear: AML blasts SU353 (light blue), AML blasts SU070 (dark blue), AML LSC SU353 (light green), AML LSC SU070 (darkgreen).

In Figure 4b/c, we have tested multiple combinations of colors, alpha parameter for transparency and size, and didn't find a truly better way of representing the points in two dimensions so that the points are neither too small, too transparent or too hard to distinguish from one another.

4. Fig. 4a: "Count" is ambiguous. The label should be replaced by "Read count", "Number of mapped reads", or something similar.

We have modified Figure 4a legend to 'Read count'.

5. Fig. 4d: No color key is provided.

We have now added a clear color key.

6. Fig. 4f: Not clear what "Count" means in these plots.

We have now modified Figure 4f legend to specify 'Read Count'.

7. Fig. 4h: Not entirely clear how "top 15" pathways were ranked – by adjusted p-value? The plot would probably be easier to read if rotated clockwise.

We have now specified that the Top 15 pathways were ranked by adjusted p-value and rotated the plot clockwise.

8. "Single-cell H3K27me3 enrichment levels can be visualized for genes within the top 100 most significant differential regions using a t-SNE." (l. 188ff) – I don't understand how a t-SNE (low-dimensional projection) is used for visualization of enrichments. Does this refer to overlaying H3K27me3 counts at selected genes onto a t-SNE plot, as in Fig. 4f?

Indeed, the plot represents the overlay of H3K27me3 counts at selected loci on to the t-SNE plot. We have modified our wording to clarify this point: "For the top 100 most significant differential regions, single-cell H3K27me3 enrichment levels can be visualized overlaying H3K27me3 counts for each cell at selected genes onto a t-SNE plot".

9. I note a competing interests statement has been added without disclosure of the names of all parties involved. I'm not sure what the journal's policy is on this.

We have added names of involved parties.

10. Support for multiple data inputs has been provided. Since data files can become quite large, a straightforward improvement (I would argue a necessary improvement) would be to add support for gzipped files (with a gzip connection wrapper). They can be largely handled in the same way as uncompressed text files but save on disk space and data transfer.

This is a very good and straightforward improvement that we will implement on the next bioconductor release (Early 2021) for .txt, .tsv and BED files. We are also planning on accepting the '.mtx' widely used format.

11. *Why is there a maximum upload size even when I'm trying to upload to my local server? I couldn't upload data obtained from Satpathy et al 2019 (GSE129785).*

Indeed, the upload limit was set by default to 5GB, we have increased the upload limit to 50Gb which should be sufficient for most datasets.

12. *I did not manage to select files using the multi-file selector for "Index, Peak & Barcode files", "Single-cell BAM files", or "Single-cell BED files" inputs when using Google Chrome browser: I get a popup dialog and can browser files, but I did not find a way to select files. Perhaps I missed something (if so, can it be made more obvious?). Why not just use the standard file selector as for "Count matrix(ces)"?*

The button was misleading and we have clarified that the user is supposed to select a folder containing all the files and not directly the files. We reasoned that selecting thousands of files could be tedious, especially as the file browser tends to be unresponsive at this high number of files. The labels now clearly indicate "Upload folder (BED/BAM/Peak-Index-Barcode)" and "Select folder containing raw files". We also updated the help associated to the corresponding help button.

*Finally an idea for future improvements: The authors perform pathway enrichment analysis using MSigDB. There are a number of tools for performing functional enrichment analysis either based on associating genomic regions with genes (e.g., GREAT, McLean et al (2010), <http://great.stanford.edu/>) or based on overlaps with "known" genomic regions (e.g. coloc-stats, Simovski et al (2018), <https://hyperbrowser.uio.no/coloc-stats/>; LolaWeb, Nagraj et al (2018), <http://lolaweb.databio.org/>). The authors may consider inclusion / linking out to these tools (many also come as web services) as a future improvement for ChromSCape.*

We thank the reviewer for these very relevant ideas of improvement. We will add such tools in a future Bioconductor release and we will follow the reviewer's recommendation so that users can choose between multiple tools for Gene Set Analysis.